# RECOMBINER: ROBUST AND ENHANCED COMPRESSION WITH BAYESIAN IMPLICIT NEURAL REPRESENTATIONS

**Jiajun He**\*
University of Cambridge
jh2383@cam.ac.uk

**Gergely Flamich**\*
University of Cambridge
gf332@cam.ac.uk

**Zongyu Guo**
University of Science and Technology of China
guozy@mail.ustc.edu.cn

**José Miguel Hernández-Lobato**
University of Cambridge
jmh233@cam.ac.uk

## ABSTRACT

COMpression with Bayesian Implicit NEural Representations (COMBINER) is a recent data compression method that addresses a key inefficiency of previous Implicit Neural Representation (INR)-based approaches: it avoids quantization and enables direct optimization of the rate-distortion performance. However, COMBINER still has significant limitations: 1) it uses factorized priors and posterior approximations that lack flexibility; 2) it cannot effectively adapt to local deviations from global patterns in the data; and 3) its performance can be susceptible to modeling choices and the variational parameters' initializations. Our proposed method, Robust and Enhanced COMBINER (RECOMBINER), addresses these issues by 1) enriching the variational approximation while retaining a low computational cost via a linear reparameterization of the INR weights, 2) augmenting our INRs with learnable positional encodings that enable them to adapt to local details and 3) splitting high-resolution data into patches to increase robustness and utilizing expressive hierarchical priors to capture dependency across patches. We conduct extensive experiments across several data modalities, showcasing that RECOMBINER achieves competitive results with the best INR-based methods and even outperforms autoencoder-based codecs on low-resolution images at low bitrates. Our PyTorch implementation is available at https://github.com/cambridge-mlg/RECOMBINER/.

## 1 INTRODUCTION

Advances in deep learning recently enabled a new data compression technique impossible with classical approaches: we train a neural network to memorize the data (Stanley, 2007) and then encode the network's weights instead. These networks are called the *implicit neural representation* (INR) of the data, and differ from neural networks used elsewhere in three significant ways. First, they treat data as a signal that maps from coordinates to values, such as mapping $(X, Y)$ pixel coordinates to $(R, G, B)$ color triplets in the case of an image. Second, their architecture consists of many fewer layers and units than usual and tends to utilize SIREN activations (Sitzmann et al., 2020). Third, we aim to *overfit* them to the data as much as possible.

Unfortunately, most INR-based data compression methods cannot directly and jointly optimize *rate-distortion*, which results in a wasteful allocation of bits leading to suboptimal coding performance. COMpression with Bayesian Implicit NEural Representations (COMBINER; Guo et al., 2023) addresses this issue by picking a variational Gaussian mean-field Bayesian neural network (Blundell et al., 2015) as the INR of the data. This choice enables joint rate-distortion optimization via maximizing the INR's $\beta$-evidence lower bound ($\beta$-ELBO), where $\beta$ controls the rate-distortion trade-off.

---

\*equal contribution.

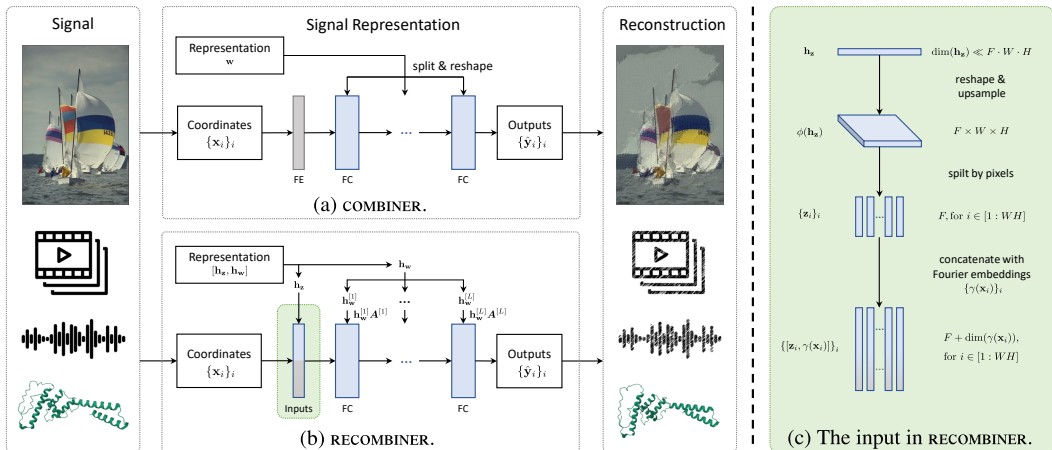

Figure 1: Schematic of (a) COMBINER and (b) RECOMBINER, our proposed method. See Sections 2 and 3 for notation. As the INR's input, RECOMBINER uses $\mathbf{h_z}$ upsampled to pixel-wise positional encodings concatenated with Fourier embeddings. (c) A closer look at how RECOMBINER maps $\mathbf{h_z}$ to the INR input, taking images as an example. **FE:** Fourier embeddings; **FC:** fully connected layer.

Finally, the authors encode a weight sample from the INR's variational weight posterior to represent the data using relative entropy coding (REC; Havasi et al., 2018; Flamich et al., 2020).

Although COMBINER performs strongly among INR-based approaches, it falls short of the state-of-the-art codecs on well-established data modalities both in terms of performance and robustness. In this paper, we identify several issues that lead to this discrepancy: 1) COMBINER employs a fully-factorized Gaussian variational posterior over the INR weights, which tends to underfit the data (Dusenberry et al., 2020), going directly against our goal of overfitting; 2) Overfitting small INRs used by COMBINER is challenging, especially at low bitrates: a small change to any weight can significantly affect the reconstruction at every coordinate, hence optimization by stochastic gradient descent becomes unstable and yields suboptimal results. 3) Overfitting becomes more problematic on high-resolution signals. As highlighted by Guo et al. (2023), the method is sensitive to model choices and the variational parameters' initialization and requires considerable effort to tune.

We tackle these problems by proposing several non-trivial extensions to COMBINER, which significantly improve the rate-distortion performance and robustness to modeling choices. Hence, we dub our method *robust and enhanced* COMBINER (RECOMBINER). Concretely, our contributions are:

- We propose a simple yet effective learned reparameterization for neural network weights specifically tailored for INR-based compression, yielding more expressive variational posteriors while matching the computational cost of standard mean-field variational inference.

- We augment our INR with learnable positional encodings whose parameters only have a local influence on the reconstructed signal, thus allowing deviations from the global patterns captured by the network weights, facilitating overfitting the INR with gradient descent.

- We split high-resolution data into patches to improve robustness to modeling choices and the variational parameters' initialization. Moreover, we propose an expressive hierarchical Bayesian model to capture the dependencies across patches to enhance performance.

- We conduct extensive experiments to verify the effectiveness of our proposed extensions across several data modalities, including image, audio, video and protein structure data. In particular, we show that RECOMBINER achieves better rate-distortion performance than VAE-based approaches on low-resolution images at low bitrates.

## 2 BACKGROUND

This section reviews the essential parts of Guo et al. (2023)'s compression with Bayesian implicit neural representations (COMBINER), as it provides the basis for our method.

**Variational Bayesian Implicit Neural Representations:** We assume the data we wish to compress can be represented as a continuous function $f : \mathbb{R}^{\mathtt{I}} \to \mathbb{R}^{\mathtt{O}}$ from $\mathtt{I}$-dimensional coordinates to $\mathtt{O}$-dimensional signal values. Then, our goal is to approximate $f$ with a small neural network $g(\cdot \mid \mathbf{w})$

with weights $\mathbf{w}$. Given $L$ hidden layers in the network, we write $\mathbf{w} = [\mathbf{w}^{[1]}, \ldots, \mathbf{w}^{[L]}]$, which represents the concatenation of the $L$ weight matrices $\mathbf{w}^{[1]}, \ldots \mathbf{w}^{[L]}$, each flattened into a row-vector. Guo et al. (2023) propose using variational Bayesian neural networks (BNN; Blundell et al., 2015) that place a prior $p_{\mathbf{w}}$ and a variational posterior $q_{\mathbf{w}}$ on the weights. Furthermore, they use Fourier embeddings $\gamma(\mathbf{x})$ for the input data (Tancik et al., 2020) and sine activations at the hidden layers (Sitzmann et al., 2020). To infer the implicit neural representation (INR) for some data $\mathcal{D}$, we treat $\mathcal{D}$ as a dataset of coordinate-value pairs $\{(\mathbf{x}_i, \mathbf{y}_i)\}_{i=1}^D$, e.g. for an image, $\mathbf{x}_i$ can be an $(X, Y)$ pixel coordinate and $\mathbf{y}_i$ the corresponding $(R, G, B)$ triplet. Next, we pick a *distortion* metric $\Delta$ (e.g., mean squared error) and a trade-off parameter $\beta$ to define the $\beta$-rate-distortion objective:

$$\mathcal{L}(\mathcal{D}, q_{\mathbf{w}}, p_{\mathbf{w}}, \beta) = \beta \cdot D_{\mathrm{KL}}[q_{\mathbf{w}} \| p_{\mathbf{w}}] + \frac{1}{D} \sum_{i=1}^D \mathbb{E}_{q_{\mathbf{w}}} \left[ \Delta(\mathbf{y}_i, g(\mathbf{x}_i \mid \mathbf{w})) \right], \tag{1}$$

where $D_{\mathrm{KL}}[q_{\mathbf{w}} \| p_{\mathbf{w}}]$ denotes the Kullback-Leibler divergence of $q_{\mathbf{w}}$ from $p_{\mathbf{w}}$, and as we explain below, it represents the compression rate of a single weight sample $\mathbf{w} \sim q_{\mathbf{w}}$. Note that Equation (1) corresponds to a negative $\beta$-evidence lower bound under mild assumptions on $\Delta$.

We infer the optimal posterior by computing $q_{\mathbf{w}}^* = \arg \min_{q_{\mathbf{w}} \in \mathcal{Q}} \mathcal{L}(\mathcal{D}, q_{\mathbf{w}}, p_{\mathbf{w}}, \beta)$ over an appropriate variational family $\mathcal{Q}$. Guo et al. (2023) set $\mathcal{Q}$ to be the family of factorized Gaussian distributions.

**Training COMBINER:** Once we selected a network architecture $g$ for our INRs, a crucial element of COMBINER is to select a good prior on the weights $p_{\mathbf{w}}$. Given a training set $\{\mathcal{D}_1, \ldots, \mathcal{D}_M\}$ and an initial guess for $p_{\mathbf{w}}$, Guo et al. (2023) propose the following iterative scheme to select the optimal prior: 1) Fix $p_{\mathbf{w}}$ and infer the variational INR posteriors $q_{\mathbf{w},m}^*$ for each datum $\mathcal{D}_m$ by minimizng Equation (1); 2) Fix the $q_{\mathbf{w},m}^*$s and update the prior parameters $p_{\mathbf{w}}$ based on the parameters of the posteriors. When the $q_{\mathbf{w}}$ are Gaussian, Guo et al. (2023) derive analytic formulae for updating the prior parameters. To avoid overloading the notion of training, we refer to learning $p_{\mathbf{w}}$ and the other model parameters as *training*, and to learning $q_{\mathbf{w}}$ as *inferring* the INR.

**Compressing data with COMBINER:** Once we picked the INR architecture $g$ and found the optimal prior $p_{\mathbf{w}}$, we can use COMBINER to compress new data $\mathcal{D}$ in two steps: 1) We first infer the variational INR posterior $q_{\mathbf{w}}$ for $\mathcal{D}$ by optimizing Equation (1), after which 2) we encode an approximate sample from $q_{\mathbf{w}}$ using relative entropy coding (REC), whose expected coding cost is approximately $D_{\mathrm{KL}}[q_{\mathbf{w}} \| p_{\mathbf{w}}]$ (Havasi et al., 2018; Flamich et al., 2020). Following Guo et al. (2023), we used depth-limited global-bound A* coding (Flamich et al., 2022), to which we will refer as just A* coding. Unfortunately, applying A* coding to encode a sample from $q_{\mathbf{w}}$ is infeasible in practice, as the time complexity of the algorithm grows as $\Omega(\exp(D_{\mathrm{KL}}[q_{\mathbf{w}} \| p_{\mathbf{w}}]))$. Hence, Guo et al. (2023) suggest breaking up the problem into smaller ones. First, they draw a uniformly random permutation $\alpha$ on $\dim(\mathbf{w})$ elements, and use it to permute the dimensions of $\mathbf{w}$ as $\alpha(\mathbf{w}) = [\mathbf{w}_{\alpha(1)}, \ldots, \mathbf{w}_{\alpha(\dim(\mathbf{w}))}]$. Then, they partition $\alpha(\mathbf{w})$ into smaller *blocks*, and compress the blocks sequentially. Permuting the weight vector ensures that the KL divergences are spread approximately evenly across the blocks. As an additional technical note, between compressing each block, we run a few steps of finetuning the posterior of the weights that are yet to be compressed, see Guo et al. (2023) for more details.

## 3 METHODS

In this section, we propose several extensions to Guo et al. (2023)'s framework that significantly improve its robustness and performance: 1) we introduce a linear reparemeterization for the INR's weights which yields a richer variational posterior family; 2) we augment the INR's input with learned positional encodings to capture local features in the data and to assist overfitting; 3) we scale our method to high-resolution image compression by dividing the images into patches and introducing an expressive hierarchical Bayesian model over the patch-INRs, and 4) we introduce minor modifications to the training procedure and adaptively select $\beta$ to achieve the desired coding budget. Contributions 1) and 2) are depicted in Figure 1, while 3) is shown in Figure 2.

### 3.1 LINEAR REPARAMETERIZATION FOR THE NETWORK PARAMETERS

A significant limitation of the factorized Gaussian variational posterior used by COMBINER is that it posits dimension-wise independent weights. This assumption is known to be unrealistic (Izmailov et al., 2021) and to underfit the data (Dusenberry et al., 2020), which goes directly against our goal of overfitting the data. On the other hand, using a full-covariance Gaussian posterior approximation would increase the INR's training and coding time significantly, even for small network architectures.

Hence, we propose a solution that lies in-between: at a high level, we learn a linearly-transformed factorized Gaussian approximation that closely matches the full-covariance Gaussian posterior on average over the training data. Formally, for each layer $l = 1, \dots, L$, we model the weights as $\mathbf{w}^{[l]} = \mathbf{h}_\mathbf{w}^{[l]} \boldsymbol{A}^{[l]}$, where the $\boldsymbol{A}^{[l]}$ are square matrices, and we place a factorized Gaussian prior and variational posterior on $\mathbf{h}_\mathbf{w}^{[l]}$ instead. We learn each $\boldsymbol{A}^{[l]}$ during the training stage, after which we fix them and only infer factorized posteriors $q_{\mathbf{h}_\mathbf{w}^{[l]}}$ when compressing new data. To simplify notation, we collect the $\boldsymbol{A}^{[l]}$ in a block-diagonal matrix $\boldsymbol{A} = \mathrm{diag}(\boldsymbol{A}^{[1]}, \dots, \boldsymbol{A}^{[L]})$ and the $\mathbf{h}_\mathbf{w}^{[l]}$ in a single row-vector $\mathbf{h}_\mathbf{w} = [\mathbf{h}_\mathbf{w}^{[1]}, \dots, \mathbf{h}_\mathbf{w}^{[L]}]$, so that now the weights are given by $\mathbf{w} = \mathbf{h}_\mathbf{w} \boldsymbol{A}$. We found this layer-wise weight reparameterization as efficient as using a joint one for the entire weight vector $\mathbf{w}$. Hence, we use the layer-wise approach, as it is more parameter and compute-efficient.

This simple yet expressive variational approximation has a couple of advantages. First, it provides an expressive full-covariance prior and posterior while requiring much less training and coding time. Specifically, the KL divergence required by Equation (1) is still between factorized Gaussians and we do not need to optimize the full covariance matrices of the posteriors during coding. Second, this parameterization has scale redundancy: for any $c \in \mathbb{R}$ we have $\mathbf{h}_\mathbf{w} \boldsymbol{A} = (1/c \cdot \mathbf{h}_\mathbf{w})(c \cdot \boldsymbol{A})$. Hence, if we initialize $\mathbf{h}_\mathbf{w}$ suboptimally during training, $\boldsymbol{A}$ can still learn to compensate for it, making our method more robust. Finally, note that this reparameterization is specifically tailored for INR-based compression and would usually not be feasible in other BNN use-cases, since we learn $\boldsymbol{A}$ while inferring multiple variational posteriors simultaneously.

## 3.2 LEARNED POSITIONAL ENCODINGS

A challenge for overfitting INRs, especially at low bitrates is their *global* representation of the data, in the sense that each of their weights influences the reconstruction at every coordinate. To mitigate this issue, we extend our INRs to take a learned positional input $\mathbf{z}_i$ at each coordinate $\mathbf{x}_i$: $g(\mathbf{x}_i, \mathbf{z}_i \mid \mathbf{w})$.

However, it is usually wasteful to introduce a vector for each coordinate in practice. Instead, we use a lower-dimensional row-vector representation $\mathbf{h}_\mathbf{z}$, that we reshape and upsample with a learnable function $\phi$. In the case of a $W \times H$ image with $F$-dimensional positional encodings, we could pick $\mathbf{h}_\mathbf{z}$ such that $\dim(\mathbf{h}_\mathbf{z}) \ll F \cdot W \cdot H$, then reshape and upsample it to be $F \times W \times H$ by picking $\phi$ to be some small convolutional network. Then, we set $\mathbf{z}_i = \phi(\mathbf{h}_\mathbf{z})_{\mathbf{x}_i}$ to be the positional encoding at location $\mathbf{x}_i$. We placed a factorized Gaussian prior and variational posterior on $\mathbf{h}_\mathbf{z}$. Hereafter, we refer to $\mathbf{h}_\mathbf{z}$ as the *latent* positional encodings, $\phi(\mathbf{h}_\mathbf{z})$ and $\mathbf{z}_i$ as the *upsampled* positional encodings.

## 3.3 SCALING TO HIGH-RESOLUTION DATA WITH PATCHES

With considerable effort, Guo et al. (2023) successfully scaled COMBINER to high-resolution images by significantly increasing the number of INR parameters. However, they note that the training procedure was very sensitive to hyperparameters, including the initialization of variational parameters and model size selection. Unfortunately, improving the robustness of large INRs using the weight reparameterization we describe in Section 3.1 is also impractical, because the size of the transformation matrix $\boldsymbol{A}$ grows quadratically in the number of weights. Therefore, we split high-resolution data into patches and infer a separate small INR for each patch, in line with other INR-based works as well (Dupont et al., 2022; Schwarz & Teh, 2022; Schwarz et al., 2023). However, the patches' INRs are independent by default, hence we re-introduce information sharing between the patch-INRs' weights via a hierarchical model for $\mathbf{h}_\mathbf{w}$. Finally, we take advantage of the patch structure to parallelize data compression and reduce the encoding time in RECOMBINER, as discussed at the end of this section.

**RECOMBINER's hierarchical Bayesian model:** We posit a global representation for the weights $\overline{\mathbf{h}}_\mathbf{w}$, from which each patch-INR can deviate. Thus, assuming that the data $\mathcal{D}$ is split into $P$ patches, for each patch $\pi \in 1, \dots, P$, we need to define the conditional distributions of patch representations $\mathbf{h}_\mathbf{w}^{(\pi)} \mid \overline{\mathbf{h}}_\mathbf{w}$. However, since we wish to model deviations from the global representation, it is natural to decompose the patch representation as $\mathbf{h}_\mathbf{w}^{(\pi)} = \Delta \mathbf{h}_\mathbf{w}^{(\pi)} + \overline{\mathbf{h}}_\mathbf{w}$, and specify the conditional distribution of the differences $\Delta \mathbf{h}_\mathbf{w}^{(\pi)} \mid \overline{\mathbf{h}}_\mathbf{w}$ instead, without any loss of generality. In this paper, we place a factorized Gaussian prior and variational posterior on the joint distribution of the global

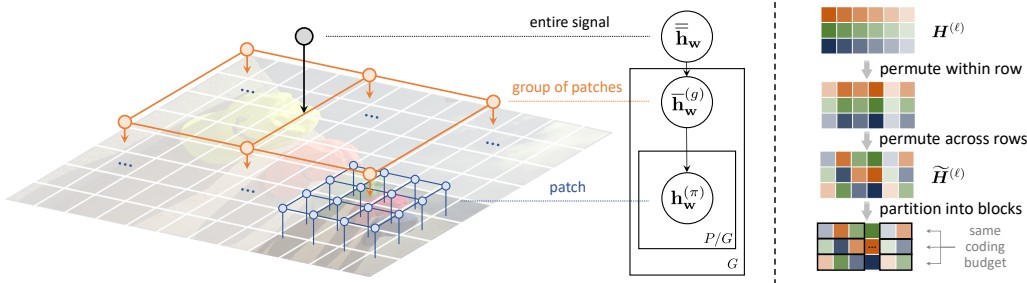

(a) Three-level hierarchical model and the corresponding graphical model.  (b) Permutation.

Figure 2: Illustration of (a) the three-level hierarchical model and (b) our permutation strategy.

representation and the deviations, given by the following product of $P + 1$ Gaussian measures:

$$p_{\overline{\mathbf{h}}_{\mathbf{w}}, \Delta \mathbf{h}_{\mathbf{w}}^{(1:P)}} = \mathcal{N}(\overline{\boldsymbol{\mu}}_{\mathbf{w}}, \mathrm{diag}(\overline{\boldsymbol{\sigma}}_{\mathbf{w}})) \times \prod_{\pi=1}^{P} \mathcal{N}(\boldsymbol{\mu}_{\Delta}^{(\pi)}, \mathrm{diag}(\boldsymbol{\sigma}_{\Delta}^{(\pi)})) \tag{2}$$

$$q_{\overline{\mathbf{h}}_{\mathbf{w}}, \Delta \mathbf{h}_{\mathbf{w}}^{(1:P)}} = \mathcal{N}(\overline{\boldsymbol{\nu}}_{\mathbf{w}}, \mathrm{diag}(\overline{\boldsymbol{\rho}}_{\mathbf{w}})) \times \prod_{\pi=1}^{P} \mathcal{N}(\boldsymbol{\nu}_{\Delta}^{(\pi)}, \mathrm{diag}(\boldsymbol{\rho}_{\Delta}^{(\pi)})), \tag{3}$$

where $1 : P$ is the slice notation, i.e. $\Delta \mathbf{h}_{\mathbf{w}}^{(1:P)} = \Delta \mathbf{h}_{\mathbf{w}}^{(1)}, \ldots, \Delta \mathbf{h}_{\mathbf{w}}^{(P)}$. Importantly, while the posterior approximation in Equation (3) assumes that the global representation and the differences are independent, $\overline{\mathbf{h}}_{\mathbf{w}}$ and $\mathbf{h}_{\mathbf{w}}^{(\pi)}$ remain correlated. Note that optimizing Equation (1) requires us to compute $D_{\mathrm{KL}}[q_{\mathbf{h}_{\mathbf{w}}^{(1:P)}} \| p_{\mathbf{h}_{\mathbf{w}}^{(1:P)}}]$. Unfortunately, due to the complex dependence between the $\mathbf{h}_{\mathbf{w}}^{(\pi)}$s, this calculation is infeasible. Instead, we can minimize an *upper bound* to it by observing that

$$\begin{aligned} D_{\mathrm{KL}}[q_{\mathbf{h}_{\mathbf{w}}^{(1:P)}} \| p_{\mathbf{h}_{\mathbf{w}}^{(1:P)}}] &\leqslant D_{\mathrm{KL}}[q_{\mathbf{h}_{\mathbf{w}}^{(1:P)}} \| p_{\mathbf{h}_{\mathbf{w}}^{(1:P)}}] + D_{\mathrm{KL}}[q_{\overline{\mathbf{h}}_{\mathbf{w}} | \mathbf{h}_{\mathbf{w}}^{(1:P)}} \| p_{\overline{\mathbf{h}}_{\mathbf{w}} | \mathbf{h}_{\mathbf{w}}^{(1:P)}}] \\ &= D_{\mathrm{KL}}[q_{\overline{\mathbf{h}}_{\mathbf{w}}, \mathbf{h}_{\mathbf{w}}^{(1:P)}} \| p_{\overline{\mathbf{h}}_{\mathbf{w}}, \mathbf{h}_{\mathbf{w}}^{(1:P)}}] \\ &= D_{\mathrm{KL}}[q_{\overline{\mathbf{h}}_{\mathbf{w}}, \Delta \mathbf{h}_{\mathbf{w}}^{(1:P)}} \| p_{\overline{\mathbf{h}}_{\mathbf{w}}, \Delta \mathbf{h}_{\mathbf{w}}^{(1:P)}}]. \end{aligned} \tag{4}$$

Hence, when training the patch-INRs, we replace the KL term in Equation (1) with the divergence in Equation (4), which is between factorized Gaussian distributions and cheap to compute. Finally, we remark that we can view $\overline{\mathbf{h}}_{\mathbf{w}}$ as side information also prevalent in other neural compression codecs (Ballé et al., 2018), or auxiliary latent variables enabling factorization (Koller & Friedman, 2009).

While Equations (2) and (3) describe a two-level hierarchical model, we can easily extend the hierarchical structure by breaking up patches further into sub-patches and adding extra levels to the probabilistic model. For our experiments on high-resolution audio, images, and video, we found that a three-level hierarchical model worked best, with global weight representation $\overline{\overline{\mathbf{h}}}_{\mathbf{w}}$, second/group-level representations $\overline{\mathbf{h}}_{\mathbf{w}}^{(1:G)}$ and third/patch-level representations $\mathbf{h}_{\mathbf{w}}^{(1:P)}$, illustrated in Figure 2a. Empirically, a hierarchical model for $\mathbf{h}_{\mathbf{z}}$ did not yield significant gains, thus we only use it for $\mathbf{h}_{\mathbf{w}}$.

**Compressing high-resolution data with RECOMBINER:** An advantage of patching is that we can compress and fine-tune INRs and latent positional encodings of all patches in parallel. Unfortunately, compressing $P$ patches in parallel using COMBINER's procedure is suboptimal, since the information content between patches might vary significantly. However, by carefully permuting the weights *across* the patches' representations we can 1) adaptively allocate bits to each patch to compensate for the differences in their information content and 2) enforce the same coding budget across each parallel thread to ensure consistent coding times. Concretely, we stack representations of each patch in a matrix at each level of the hierarchical model. For example, in our three-level model we set

$$\boldsymbol{H}_{\pi,:}^{(0)} = [\mathbf{h}_{\mathbf{w}}^{(\pi)}, \mathbf{h}_{\mathbf{z}}^{(\pi)}], \quad \boldsymbol{H}_{g,:}^{(1)} = \overline{\mathbf{h}}_{\mathbf{w}}^{(g)}, \quad \boldsymbol{H}^{(2)} = \overline{\overline{\mathbf{h}}}_{\mathbf{w}}, \tag{5}$$

where we use slice notation to denote the $i$th row as $\boldsymbol{H}_{i,:}$ and the $j$th column as $\boldsymbol{H}_{:,j}$. Furthermore, let $S_n$ denote the set of permutations on $n$ elements. Now, at each level $\ell$, assume $\boldsymbol{H}^{(\ell)}$ has $\mathcal{C}_\ell$ columns and $\mathcal{R}_\ell$ rows. We sample a single within-row permutation $\kappa$ uniformly from $S_{\mathcal{C}_\ell}$ and for each column of $\boldsymbol{H}^{(\ell)}$ we sample an across-rows permutation $\alpha_j$ uniformly from $S_{\mathcal{R}_\ell}$ elements. Then, we permute $\boldsymbol{H}^{(\ell)}$ as $\widetilde{\boldsymbol{H}}_{i,j}^{(\ell)} = \boldsymbol{H}_{\alpha_j(i),\kappa(j)}^{(\ell)}$. Finally, we split the $\boldsymbol{H}^{(\ell)}$s into blocks row-wise, and encode and fine-tune each row in parallel. We illustrate the above procedure in Figure 2b.

## 3.4 EXTENDED TRAINING PROCEDURE

In this section, we describe the ways in which RECOMBINER's training procedure deviates from COMBINER's. To begin, we collect the RECOMBINER's representations into one vector. For non-patching cases we set $\mathbf{h} = [\mathbf{h_w}, \mathbf{h_z}]$, and for the patch case using the three-level hierarchical model we set $\mathbf{h} = \text{vec}([\boldsymbol{H}^{(0)}, \boldsymbol{H}^{(1)}, \boldsymbol{H}^{(2)}])$. For simplicity, we denote the factorized Gaussian prior and variational posterior over $\mathbf{h}$ as $p_{\mathbf{h}} = \mathcal{N}(\boldsymbol{\mu}, \text{diag}(\boldsymbol{\sigma}))$ and $q_{\mathbf{h}} = \mathcal{N}(\boldsymbol{\nu}, \text{diag}(\boldsymbol{\rho}))$, where $\boldsymbol{\mu}$ and $\boldsymbol{\nu}$ are the means and $\boldsymbol{\sigma}$ and $\boldsymbol{\rho}$ are the diagonals of covariances of the prior and the posterior, respectively.

**Training RECOMBINER:** Our objective for the training stage is to obtain the model parameters $\boldsymbol{A}, \phi, \boldsymbol{\mu}, \boldsymbol{\sigma}$ given a training dataset $\{\mathcal{D}_1, \ldots, \mathcal{D}_M\}$ and a coding budget $C$. [1] In their work, Guo et al. (2023) control the coding budget *implicitly* by manually setting different values for $\beta$ in Equation (1). In this paper, we adopt an *explicit* approach and tune $\beta$ dynamically based on our desired coding budget of $C$ bits. More precisely, after every iteration, we calculate the average KL divergence of the training examples, i.e., $\bar{\delta} = \frac{1}{M} \sum_{m=1}^{M} D_{\text{KL}}[q_{\mathbf{h},m} || p_{\mathbf{h}}]$. If $\bar{\delta} > C$, we update $\beta$ by $\beta \leftarrow \beta \times (1 + \tau_C)$; if $\bar{\delta} < C - \epsilon_C$, we update $\beta$ by $\beta \leftarrow \beta / (1 + \tau_C)$. Here $\epsilon_C$ is a threshold parameter to stabilize the training process and prevent overly frequent updates to $\beta$, and $\tau_C$ is the adjustment step size. Unless otherwise stated, we set $\tau_C = 0.5$ in our experiments. Empirically, we find the value of $\beta$ stabilizes after 30 to 50 iterations. We present the pseudocode of this prior learning algorithm in Algorithm 1. Then, our training step is a three-step coordinate descent process analogous to Guo et al. (2023)'s:

1. **Optimize variational parameters, linear transformation and upsampling network:** Fix the prior $p_{\mathbf{h}}$, and optimize Equation (1) or its modified version from Section 3.3 via gradient descent. Note, that $\mathcal{L}$ is a function of the linear transform $\boldsymbol{A}$ and upsampling network parameters $\phi$ too:

$$\{\boldsymbol{\nu}_m, \boldsymbol{\rho}_m\}_{m=1}^{M}, \boldsymbol{A}, \phi \quad \leftarrow \quad \underset{\{\boldsymbol{\nu}_m, \boldsymbol{\rho}_m\}_{m=1}^{M}, \boldsymbol{A}, \phi}{\arg\min} \left\{ \frac{1}{M} \sum_{m=1}^{M} \mathcal{L}(\mathcal{D}_m, q_{\mathbf{h},m}, p_{\mathbf{h}}, \boldsymbol{A}, \phi, \beta) \right\}. \quad (6)$$

2. **Update prior:** Update the prior parameters by the closed-form solution:

$$\boldsymbol{\mu} \leftarrow \frac{1}{M} \sum_{m=1}^{M} \boldsymbol{\nu}_m, \quad \boldsymbol{\sigma} \leftarrow \frac{1}{M} \sum_{m=1}^{M} \left[ (\boldsymbol{\nu}_m - \boldsymbol{\mu})^2 + \boldsymbol{\rho}_m \right]. \quad (7)$$

3. **Update $\beta$:** Set $\beta \leftarrow \beta \times (1 + \tau_C)$ or $\beta \leftarrow \beta / (1 + \tau_C)$ based on the procedure described above.

Note that unlike other INR-based methods (Dupont et al., 2022; Schwarz & Teh, 2022; Schwarz et al., 2023) our training procedure is remarkably stable, as we illustrate in Appendix D.4.

## 4 RELATED WORKS

**Nonlinear transform coding:** Currently, the dominant paradigm in neural compression is nonlinear transform coding (NTC; Ballé et al., 2020) usually implemented using variational autoencoders (VAE). NTC has achieved impressive performance in terms of both objective metrics (Cheng et al., 2020; He et al., 2022) and perceptual quality (Mentzer et al., 2020), mainly due to their expressive learned non-linear transforms (Ballé et al., 2020; Zhu et al., 2021; Liu et al., 2023) and elaborate entropy models (Ballé et al., 2018; Minnen et al., 2018; Guo et al., 2021).

Compressing INRs can also be viewed as a form of NTC: we use gradient descent to transform data into an INR. The idea to quantize INR weights and entropy code them was first proposed by Dupont et al. (2021), whose method has since been extended significantly (Dupont et al., 2022; Schwarz & Teh, 2022; Schwarz et al., 2023). The current state-of-the-art INR-based method, VC-INR (Schwarz et al., 2023), achieves impressive results across several data modalities, albeit at the cost of significantly higher complexity and still falling short of autoencoder-based NTC methods on images. Our method, following COMBINER (Guo et al., 2023), differs from all of the above methods, as it uses REC to encode our variational INRs, instead of quantization and entropy coding.

**Linear weight reparameterization:** Similar to our proposal in Section 3.1, Oktay et al. (2019) learn an affine reparameterization of the weights of large neural networks. They demonstrate that scalar quantization in the transformed space leads to significant gains in compression performance. However, since they are performing one-shot model compression, their linear transformations have

---

[1]As a slight abuse of notation, we use $\phi$ to denote both the upsampling function and its parameters.

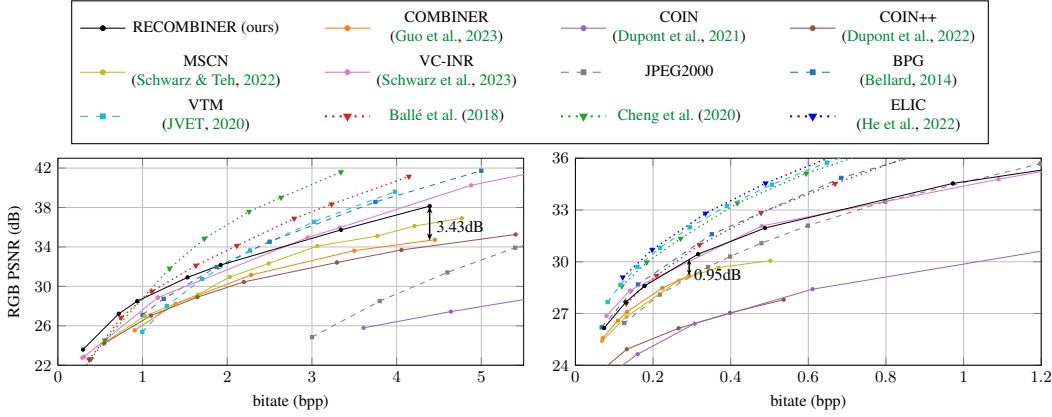

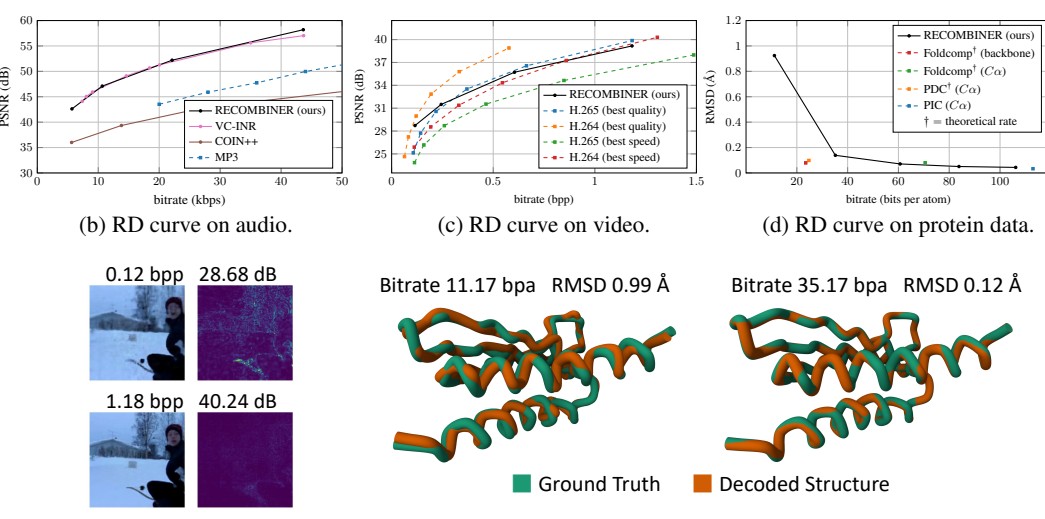

(a) RD curve on CIFAR-10 (left) and Kodak (right). We also provide full-resolution plots in Appendix F.

(b) RD curve on audio.  (c) RD curve on video.  (d) RD curve on protein data.

(e) Decoded videos and residuals.  (f) Decoded protein structure examples.

Figure 3: Quantitive evaluation and qualitative examples of RECOMBINER on image, audio, video, and 3D protein structure. Kbps stands for kilobits per second, RMSD stands for Root Mean Square Deviation, and bpa stands for bits per atom. For all plots, we use solid lines to denote INR-based codecs, dotted lines to denote VAE-based codecs, and dashed lines to denote classical codecs.

very few parameters as they need to transmit them alongside the quantized weights, limiting their expressivity. On the other hand, RECOMBINER learns the linear transform during training after which it is fixed and shared between communicating parties, thus it does not cause any communication overhead. Therefore, our linear transformation can be significantly more expressive.

**Positional encodings:** Some recent works have demonstrated that learning positional features is beneficial for fitting INRs (Jiang et al., 2020; Kim et al., 2022; Müller et al., 2022; Ladune et al., 2023). Sharing a similar motivation, our method essentially incorporates implicit representations with explicit ones, forming a hybrid INR framework (Chen et al., 2023).

## 5 EXPERIMENTAL RESULTS

In this section, we evaluate RECOMBINER on image, audio, video, and 3D protein structure data and demonstrate that it achieves strong performance across all modalities. We also perform extensive ablation studies on the CIFAR-10 and Kodak datasets which demonstrate RECOMBINER's robustness and the effectiveness of each of our proposed solutions. For all experiments, we use a 4-layer, 32-hidden unit SIREN network (Sitzmann et al., 2020) as the INR architecture unless otherwise stated, and a small 3-layer convolution network as the upsampling network $\phi$, as shown in Figure 6 in the appendix. See Appendix C for the detailed description of our experimental setup.

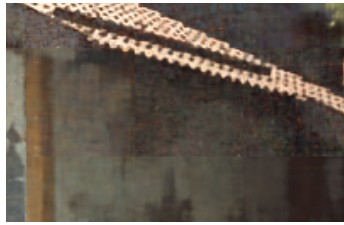 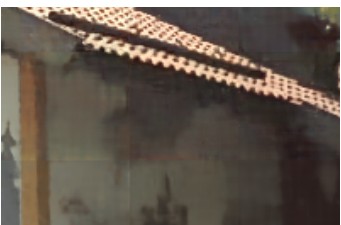 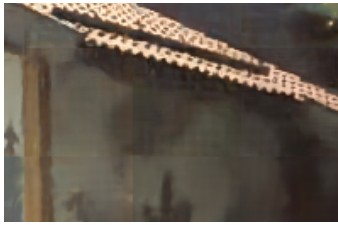

(a) w/o positional encodings;
bitrate 0.287 bpp; PSNR 25.62 dB.

(b) with positional encodings;
bitrate 0.316 bpp; PSNR 26.85 dB.

(c) with positional encodings;
bitrate 0.178 bpp; PSNR 25.05 dB.

Figure 4: Comparison between `kodim24` details compressed with and without learnable positional encodings. (a)(b) have similar bitrates and (a)(c) have similar PSNRs.

## 5.1 DATA COMPRESSION ACROSS MODALITIES

**Image:** We evaluate RECOMBINER on the CIFAR-10 (Krizhevsky et al., 2009) and Kodak (Kodak, 1993) image datasets, and show its rate-distortion (RD) performance in Figure 3a, and compare it against recent INR and VAE-based methods, as well as VTM (JVET, 2020)[2], BPG (Bellard, 2014) and JPEG2000. RECOMBINER displays remarkable performance on CIFAR-10, especially at low bitrates, outperforming even VAE-based codecs. On Kodak, it outperforms most INR-based codecs and is competitive with the more complex VC-INR method of Schwarz et al. (2023). Finally, while RECOMBINER still falls behind VAE-based codecs, it significantly reduces the performance gap.

**Audio:** Following the experimental set-up of Guo et al. (2023), we evaluate our method on the LibriSpeech (Panayotov et al., 2015) dataset. In Figure 3b, we depict RECOMBINER's RD curve on the full test set, alongside the curves of VC-INR, COIN++, and MP3. We can see RECOMBINER outperforms both COIN++ and MP3 and matches with VC-INR. Since Guo et al. (2023) only tested COMBINER on 24 test clips, we do not include COMBINER in this plot but put an extra comparison in Figure 13 in Appendix F, where we can also see that RECOMBINER clearly outperforms COMBINER.

**Video:** We evaluate RECOMBINER on UCF-101 action recognition dataset (Soomro et al., 2012), following Schwarz et al. (2023)'s experimental setup. However, as they do not report their train-test split and due to the time-consuming encoding process of our approach, we only benchmark our method against H.264 and H.265 on 16 randomly selected video clips. Figure 3c shows RECOMBINER achieves comparable performance to the classic domain-specific codecs H.264 and H.265, especially at lower bitrates. However, there is still a gap between our approach and H.264 and H.265 when they are configured to prioritize quality. Figure 3e shows a non-cherry-picked video compressed with RECOMBINER at two different bitrates and its reconstruction errors.

**3D Protein Structure:** To further illustrate the applicability of our approach, we use it to compress the 3D coordinates of $C\alpha$ atoms in protein fragments. We take domain-specific lossy codecs as baselines, including Foldcomp (Kim et al., 2023), PDC (Zhang & Pyle, 2023) and PIC (Staniscia & Yu, 2023). Surprisingly, as shown in Figure 3d, RECOMBINER's performance is competitive with highly domain-specific codecs. Furthermore, it allows us to tune its rate-distortion performance, whereas the baselines only support a certain compression rate. Since the experimental resolution of 3D structures is typically between 1-3 Å (RCSB Protein Data Bank, 2000), RECOMBINER could help with reducing the increasing storage demand for protein structures without losing key information. Figure 3f shows non-cherry-picked examples compressed with our method.

## 5.2 EFFECTIVENESS OF OUR SOLUTIONS, ABLATION STUDIES AND RUNTIME ANALYSIS

This section showcases RECOMBINER's robustness to model size and the effectiveness of each component. Appendix D.1 provides additional visualizations for a deeper understanding of our methods.

**Positional encodings facilitate local deviations:** Figure 4 compares images obtained by RECOMBINER with and without positional encodings at matching bitrates and PSNRs. As we can see, positional encodings preserve intricate details in fine-textured regions while preventing noisy artifacts in other regions of the patches, making RECOMBINER's reconstructions more visually pleasing.

---

[2]https://vcgit.hhi.fraunhofer.de/jvet/VVCSoftware_VTM/-/tree/VTM-12.0?ref_type=tags

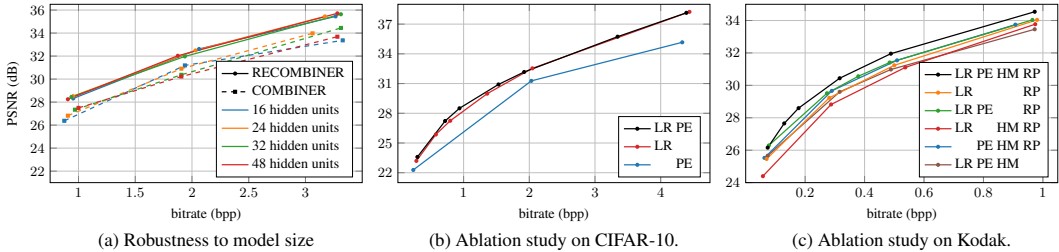

(a) Robustness to model size     (b) Ablation study on CIFAR-10.     (c) Ablation study on Kodak.

Figure 5: (a) RD performances of COMBINER and RECOMBINER with different numbers of hidden units. (b)(c) Ablation studies on CIFAR-10 and Kodak. **LR:** linear reparameterization; **PE:** positional encodings; **HM:** hierarchical model; **RP:** random permutation across patches. We describe the details of experimental settings for ablation studies in Appendix C.3.

**RECOMBINER is more robust to model size:** Using the same INR architecture, Figure 5a shows COMBINER and RECOMBINER's RD curves as we vary the number of hidden units. RECOMBINER displays minimal performance variation and also consistently outperforms COMBINER. Based on Figure 7 in Appendix D, this phenomenon is likely due to RECOMBINER's linear weight reparameterization allowing it to more flexibly prune its weight representations.

**Ablation study:** In Figures 5b and 5c, we ablate our linear reparameterization, positional encodings, hierarchical model, and permutation strategy on CIFAR-10 and Kodak, with five key takeaways:

1. Linear weight reparameterization consistently improves performance on both datasets, yielding up to 4dB gain on CIFAR-10 at high bitrates and over 0.5 dB gain on Kodak in PSNR.

2. Learnable positional encodings provide more substantial advantages at lower bitrates. On CIFAR-10, the encodings contribute up to 0.5 dB gain when the bitrate falls below 2 bpp. On Kodak, the encodings provide noteworthy gains of 2 dB at low bitrates and 1 dB at high bitrates.

3. Surprisingly, the hierarchical model *without positional encodings* can degrade performance. We hypothesize that this is because directly applying the hierarchical model poses challenges in optimizing Equation (1). A potential solution is to warm up the rate penalty $\beta$ level by level akin to what is done in hierarchical VAEs (Sønderby et al., 2016), which we leave for further work.

4. However, positional encodings appear to consistently alleviate this optimization difficulty, yielding 0.5 dB gain when used with hierarchical models.

5. Our proposed permutation strategy provides significant gains of 0.5 dB at low bitrates and more than 1.5 dB at higher bitrates.

**Runtime Analysis:** We list RECOMBINER's encoding and decoding times in Appendix D.5. Unfortunately, our approach exhibits a long encoding time, similar to COMBINER. However, our decoding process is still remarkably fast, matching the speed of COIN and COMBINER, even on CPUs.

## 6 CONCLUSIONS AND LIMITATIONS

In this paper, we propose RECOMBINER, a new codec based on several non-trivial extensions to COMBINER, encompassing the linear reparameterization for the network weights, learnable positional encodings, and expressive hierarchical Bayesian models for high-resolution signals. Experiments demonstrate that our proposed method sets a new state-of-the-art on low-resolution images at low bitrates, and consistently delivers strong results across other data modalities.

A major limitation of our work is the encoding time complexity and tackling it should be of primary concern in future work. A possible avenue for solving this issue is to reduce the number of parameters to optimize over and switch from inference over weights to modulations using, e.g. FiLM layers (Perez et al., 2018), as is done in other INR-based works. A second limitation is that while compressing with patches enables parallelization and higher robustness, it is suboptimal as it leads to block artifacts, as can be seen in Figure 4. Third, as Guo et al. (2023) demonstrate, the approximate samples given by A* coding significantly impact the methods performance, e.g. by requiring more fine-tuning. An interesting question is whether an exact REC algorithm could be adapted to solve this issue, such as the recently developed greedy Poisson rejection sampler (Flamich, 2023).

## 7 ACKNOWLEDGEMENTS

The authors would like to thank Runsen Feng for helping us ensure that our baseline for our experiments on video compression is correctly set up. GF acknowledges funding from DeepMind. ZG acknowledges funding from the Outstanding PhD Student Program at the University of Science and Technology of China.

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

# A  NOTATIONS

We summarize the notations used in this paper in Table 1:

| Notation | Name |
|---|---|
| $\beta$ | rate penalty hyperparameter in Equation (1) |
| $C$ | coding budget |
| $\tau_C$ | step size for adjusting $\beta$ |
| $\epsilon_C$ | threshold parameter to stabilize training when adjusting $\beta$ |
| $\mathbf{w}$ | weights in INR |
| $\mathbf{x}_i$ | $i$th coordinate |
| $\mathbf{y}_i$ | $i$th signal value |
| $\mathbf{z}_i$ | RECOMBINER's upsampled positional encodings at coordinate $\mathbf{x}_i$ |
| $\mathbf{h_w}$ | RECOMBINER's latent INR weights |
| $\mathbf{h_z}$ | RECOMBINER's latent positional encodings |
| $\mathbf{h_w}^{(\pi)}$ | latent INR weights for $\pi$th patch (lowest level of the hierarchical model) |
| $\mathbf{h_z}^{(\pi)}$ | latent positional encodings for $\pi$th patch (lowest level of the hierarchical model) |
| $\overline{\mathbf{h}}_\mathbf{w}^{(g)}$ | $g$th representation in the second level of the hierarchical model |
| $\overline{\overline{\mathbf{h}}}_\mathbf{w}$ | third level representations of the hierarchical model |
| $\boldsymbol{\nu}$ | mean of the Gaussian posterior |
| $\boldsymbol{\mu}$ | mean of the Gaussian prior |
| $\boldsymbol{\rho}$ | diagonal of the covariance matrix of the Gaussian posterior |
| $\boldsymbol{\sigma}$ | diagonal of the covariance matrix of the Gaussian prior |
| $\boldsymbol{A}$ | RECOMBINER's linear transform on INR weights |
| $\boldsymbol{H}^{(\ell)}$ | matrix stacking representations in the $\ell$th level defined in Equation (5) |
| $\widetilde{\boldsymbol{H}}^{(\ell)}$ | matrix for representations in the $\ell$th level after permutation |
| $\mathcal{D}$ | a signal data point (as a dataset with coordinate-value pairs) |
| $S_n$ | set of all permutations on $n$ elements |
| $\gamma(\cdot)$ | Fourier embedding to coordinates |
| $\alpha(\cdot), \kappa(\cdot)$ | a permutation |
| $\phi(\cdot)$ | upsampling network for positional encodings |
| $g(\cdot \mid \mathbf{w})$ | INR with weights $\mathbf{w}$ |

Table 1: Notations.

# B   RECOMBINER'S TRAINING ALGORITHMS

We describe the algorithm to train RECOMBINER in Algorithm 1.

---

**Algorithm 1** Training RECOMBINER: the prior, the linear transform $\boldsymbol{A}$ and upsampling network $\phi$

---

**Require:** Training data $\{\mathcal{D}_1, ..., \mathcal{D}_M\}$; desired bitrate $C$.
   **Initialize:** $q_{\mathbf{h},m} = \mathcal{N}\left(\boldsymbol{\nu}_m, \text{diag}\left(\boldsymbol{\rho}_{\mathbf{m}}\right)\right)$ for every training instance $\mathcal{D}_m$.
   **Initialize:** $p_{\mathbf{h}} = \mathcal{N}\left(\boldsymbol{\mu}, \text{diag}\left(\boldsymbol{\sigma}\right)\right)$.
   **Initialize:** $\boldsymbol{A}, \phi$.
   **repeat until convergence**
      # Step 1: Optimize posteriors, linear reparameterization matrix, and upsampling network
      $\{\boldsymbol{\nu}_m, \boldsymbol{\rho}_m\}_{m=1}^M, \boldsymbol{A}, \phi \quad \leftarrow \quad \arg\min_{\{\boldsymbol{\nu}_m, \boldsymbol{\rho}_m\}_{m=1}^M, \boldsymbol{A}, \phi} \left\{ \frac{1}{M} \sum_{m=1}^M \mathcal{L}(\mathcal{D}_m, q_{\mathbf{h},m}, p_{\mathbf{h}}, \boldsymbol{A}, \phi, \beta) \right\}.$
      $\rhd$ Optimize by Equation (6)

      # Step 2: Update prior
      $\boldsymbol{\mu} \leftarrow \frac{1}{M} \sum_{m=1}^M \boldsymbol{\nu}_m, \quad \boldsymbol{\sigma} \leftarrow \frac{1}{M} \sum_{m=1}^M \left[ \left(\boldsymbol{\nu}_m - \boldsymbol{\mu}\right)^2 + \boldsymbol{\rho}_m \right].$   $\rhd$ Update by Equation (7)

      # Step 3: Update $\beta$
      $\bar{\delta} = \frac{1}{M} \sum_{m=1}^M D_{\text{KL}}[q_{\mathbf{h},m} || p_{\mathbf{h}}].$   $\rhd$ Calculate the average training KL
      **if** $\bar{\delta} > C$ **then**
         $\beta \leftarrow \beta \times (1 + \tau_C)$   $\rhd$ Increase $\beta$ if budget is exceeded
      **end if**
      **if** $\bar{\delta} < C - \epsilon_C$ **then**
         $\beta \leftarrow \beta / (1 + \tau_C)$   $\rhd$ Decrease $\beta$ if budget is not fully occupied
      **end if**
   **end repeat**
   **Return:** $p_{\mathbf{h}} = \mathcal{N}\left(\boldsymbol{\mu}, \text{diag}\left(\boldsymbol{\sigma}\right)\right), \boldsymbol{A}, \phi$.

---

# C   SUPPLEMENTARY EXPERIMENTAL DETAILS

## C.1   DATASETS AND MORE DETAILS ON EXPERIMENTS

In this section, we describe the dataset and our experimental settings. We depict the upsampling network we used in Figure 6 and summarize the hyperparameters for each modality in Table 2. Besides, we present details for the baselines in Appendix C.2.

Note, that as the proposed linear reparameterization yields a full-covariance Gaussian posterior over the weights in the INR, the local reparameterization trick (Kingma et al., 2015) is not applicable in RECOMBINER. Therefore, in the above experiments, when inferring the posteriors of a test signal, we employ a Monte Carlo estimator with 5 samples to estimate the expectation in $\beta$-ELBO in Equation (1). While during the training stage, we still use 1 sample. In Appendix D.3, we provide an analysis of the sample size's influence. It is worth noting that using just 1 sample during inferring does not significantly deteriorate performance, and therefore we have the flexibility to reduce the sample size when prioritizing encoding time, with marginal performance impact.

**CIFAR-10:** CIFAR-10 is a set of low-resolution images with a size of $32 \times 32$. It has a training set of 50,000 images and a test set of 10,000 images. We randomly select 15,000 images from the training set for the training stage and evaluate RD performance on all test images. we use SIREN network (Sitzmann et al., 2020) with 4 layers and 32 hidden units as the INR architecture.

**Kodak:** Kodak dataset is a commonly used image compression benchmark, containing 24 images with resolutions of either $768 \times 512$ or $512 \times 768$. In our experiments, we split each images into 96 patches with size $64 \times 64$. Lacking a standard training set, we randomly select and crop 83 images with the same size (splitting into 7,968 patches) from the DIV2K dataset (Agustsson & Timofte, 2017) as the training set. We compress each Kodak image in $64 \times 64$ patches. For each patch, we use the same INR setup as that for CIFAR-10, i.e., SIREN network (Sitzmann et al., 2020) with 4 layers and 32 hidden units. Besides, we apply a three-level hierarchical Bayesian model to Kodak

patches. The lowest level has 96 patches. Every 16 ($4 \times 4$) patches are grouped together in the second level, and in total there are 6 groups. The highest level consists of a global representation for the entire image.

**Audio:** LibriSpeech (Panayotov et al., 2015) is a speech dataset recorded at a 16kHz sampling rate. We follow the experiment settings by Guo et al. (2023), taking the first 3 seconds of every recording, corresponding to 48,000 audio samples. We compress each audio clip with 60 patches, each of which has 800 audio samples. For each patch, we use the same INR architecture as CIFAR-10 except the output of the network has only one dimension. We train RECOMBINER on 197 training instances (corresponding to 11,820 patches) and evaluate it on the test set split by Guo et al. (2023), consisting of 24 instances. We also apply a three-level hierarchical model. The lowest level consists of 60 patches. Every 4 patches are grouped together in the second level, and in total there are $60/4 = 16$ groups. The highest level consists of a global representation for the entire signal.

**Video:** UCF-101 (Soomro et al., 2012) is a dataset of human actions. It consists of 101 action classes, over 13k clips, and 27 hours of video data. We follow Schwarz et al. (2023) center-cropping each video clip to $240 \times 240 \times 24$ and then resizing them to $128 \times 128 \times 24$. Then we compress each clip with $16 \times 16 \times 24$ patches. We train RECOMBINER on 75 video clips (4,800 patches) and evaluate it on 16 randomly selected clips. For each patch, we still use the INR with 4 layers and 32 hidden units. We also apply the three-level hierarchical model. The lowest level consists of 64 patches. Every 16 $4 \times 4$ patches are grouped together in the second level, and in total, there are 4 groups. The highest level consists of a global representation for the entire clip. **3D Protein structure:** We evaluate RECOMBINER on the *Saccharomyces cerevisiae* proteome from the AlphaFold DB v4[3]. To standardize the dataset, for each protein, we take the C$\alpha$ atom of the first 96 residues (i.e., amino acids) as the target data to be compressed. The input coordinates are the indices of the C$\alpha$ atoms (varying between 1-96, and normalized between 0-1) and the outputs of INRs are their corresponding 3D coordinates. We randomly select 1,000 structures as the test set and others as the training set. We still use the same INR architecture as CIFAR-10, i.e., SIREN network with 4 layers and 32 hidden units in each layer. We use the standard MSE as the distortion measure. Note that our method can also be extended to take the fact that the 3D structure is rotation and translation invariant into account by using different losses.

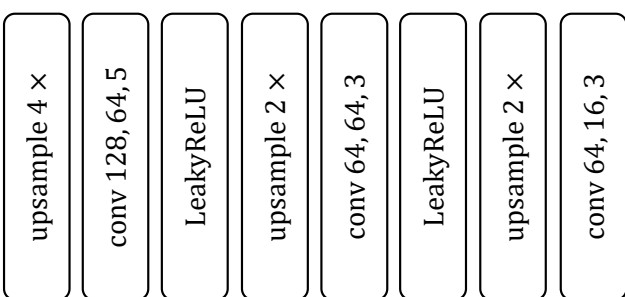

Figure 6: Architecture of the up-sampling network $\phi$ for learnable positional encodings. The numbers in the convolution layer represent the number of input channels, the number of output channels, and kernel size respectively. `same padding mode` is used in all convolution layers. The kernel dimension depends on the modality, for instances, we use kernels with sizes of 5, 3, 3 for audio and proteins, kernels with sizes of $5 \times 5$, $3 \times 3$, $3 \times 3$ for images, and kernels with sizes of $5 \times 5 \times 5$, $3 \times 3 \times 3$, $3 \times 3 \times 3$ for video.

## C.2 BASELINE SETTINGS

The baseline performances, including JPEG2000, BPG, COIN, COIN++, Ballé et al. (2018) and Cheng et al. (2020) on CIFAR-10 and Kodak, and MP3 and COIN++ on the full test set of LibriSpeech, are taken from the COIN++'s GitHub repo[4]. Statistics for VC-INR and MSCN are pro-

---

[3] https://ftp.ebi.ac.uk/pub/databases/alphafold/v4/UP000002311_559292_YEAST_v4.tar
[4] https://github.com/EmilienDupont/coinpp

| | Image | | Audio | Video | Protein |
|---|---|---|---|---|---|
| | Cifar-10 | Kodak | | | |
| **Patching** | | | | | |
| patch or not | ✗ | ✓ | ✓ | ✓ | ✗ |
| patch size | | $64 \times 64$ | 800 | $16 \times 16 \times 24$ | |
| hierarchical model levels | \ | 3 | 3 | 3 | \ |
| number of patches (lowest level) | \ | 96 | 60 | 64 | \ |
| number of groups of patches (middle level) | \ | 6 | 16 | 4 | \ |
| number of groups of groups (highest level) | \ | 1 | 1 | 1 | \ |
| **Positional Encodings** | | | | | |
| latent positional encoding shape | $128 \cdot 2 \cdot 2$ | $128 \cdot 4 \cdot 4$ | $128 \cdot 50$ | $128 \cdot 1 \cdot 1 \cdot 1$ | $128 \cdot 6$ |
| latent positional encoding param number | 512 | 2560 | 6400 | 128 | 768 |
| upsampled positional encoding shape | $16 \times 32 \times 32$ | $16 \times 64 \times 64$ | $16 \times 800$ | $16 \times 16 \times 16 \times 24$ | $16 \times 96$ |
| **INR Architecture** | | | | | |
| layers | | | 4 | | |
| hidden units | | | 32 | | |
| Fourier embeddings dimension | 16 | 16 | 16 | 18 ($\frac{16}{3}$ is not integer) | 16 |
| output dimension | 3 | 3 | 1 | 3 | 1 |
| number of parameters | 3267 | 3267 | 3201 | 3331 | 3201 |
| **Training Stage** | | | | | |
| training size | 15000 | 83 (7968 patches) | 197 (11820 patches) | 75 (4800 patches) | 4691 |
| epochs | | | 550 | | |
| optimizer | | | Adam (lr=0.0002) | | |
| sample size to estimate $\beta$-ELBO | | | 1 | | |
| gradient iteration between updating prior | | | 100 | | |
| the first gradient iteration | | | 200 | | |
| initial posterior variance | | | $9 \times 10^{-6}$ | | |
| initial posterior mean | | | SIREN initialization | | |
| initial $\boldsymbol{A}^{[l]}$ values | | | $A \sim \mathcal{U}(-1/a, 1/a), a = d_{in}d_{out}$ where $d_{in}$ and $d_{out}$ are the input and output dimension for layer $l$. | | |
| $\epsilon_C$ | 0.3 bpp | 0.05 bpp | 0.5 kbps | 0.3 bpp | 0.3 bpa |
| $\beta$ | | | Adaptively adjusted. Initial value $1 \times 10^{-8}$ | | |
| **Posterior Inferring and Compression Stage** | | | | | |
| gradient descent iteration | | | 30000 | | |
| optimizer | | | Adam (lr=0.0002) | | |
| sample size to estimate $\beta$-ELBO | | | 5 | | |
| blocks per signal (total number of blocks) | {19,46,60,98, 123,214,281} | {1819, 3187, 4373,7770, 12004, 23898} | {1066, 1999, 4146, 8182} | {2827, 5992, 14858, 29073} | {67, 211, 364 503, 637} |
| bits per block | | | 16 bits | | |
| blocks in the lowest level (patch) | \ | {17, 30, 41, 73, 114, 233} | {15, 31, 64, 122} | {34, 71, 198, 409} | \ |
| blocks in the middle level | \ | {17, 34, 52, 102, 145, 211} | {5, 5, 14, 50} | {109, 284, 427, 561} | \ |
| blocks in the highest level | \ | {85,103, 125, 150, 190, 264} | {31, 64, 96, 112} | {215, 312, 478, 653} | \ |

Table 2: Hyperparameters for images, audio, video, and protein structure compression.

vided by the authors in the paper. We also include a comparison of RECOMBINER and COMBINER on 24 test audio clips since the authors of COMBINER did not test on the full test set. For this comparison, the performances of COMBINER and MP3 on 24 test audio clips are provided by the authors of COMBINER.

Below, we describe details about the baseline of the video and protein structure compression.

### C.2.1 Video Baselines

Video compression baselines are implemented by `ffmpeg` (Tomar, 2006), with the following commands.

H.264 (best speed):

```
ffmpeg.exe -i INPUT.avi -c:v libx264 -preset ultrafast -crf $CRF
OUTPUT.mkv
```

H.265 (best speed):

```
ffmpeg.exe -i INPUT.avi -c:v libx265 -preset ultrafast -crf $CRF
OUTPUT.mkv
```

H.264 (best quality):

```
ffmpeg.exe -i INPUT.avi -c:v libx264 -preset veryslow -crf $CRF
OUTPUT.mkv
```

H.265 (best quality):

```
ffmpeg.exe -i INPUT.avi -c:v libx265 -preset veryslow -crf $CRF
OUTPUT.mkv
```

The argument `$CRF` varies in `15 20 25 30 35 40`.

### C.2.2 PROTEIN BASELINES

Softwares implementing PIC, PDC and Foldcomp are available at https://github.com/lukestaniscia/PIC, https://github.com/kad-ecoli/pdc and https://github.com/steineggerlab/foldcomp.

PIC first employs a lossy mapping, converting the 3D coordinates of atoms to an image, and then lossless compresses the image in PNG format. We use the PNG image size to calculate the bitrate.

As for PDC and Foldcomp, since they directly operate on PDB files containing other information like the headers, sequences, B factor, etc., we cannot use the file size directly. Therefore, we use their theoretical bitrates as our baseline. Below we present how we calculate their theoretical bitrates.

PDC uses three 4-byte integers to save the coordinates of the first $C\alpha$ atom, and three 1-byte integers for coordinate differences of all remaining $C\alpha$ atoms. Therefore, in theory, for a 96-residue length protein, each $C\alpha$ atom is assigned with $(8 \times 3 \times 95 + 4 \times 8 \times 3 \times 1)/96$ bits.

Foldcomp compresses the quantized dihedral/bond angles for each residue. Every residue needs 59 bits. Besides, Foldcomp saves uncompressed coordinates for every 25 residues as anchors, which requires 36 bytes. Therefore, the theoretical number of bits assigned to each $C\alpha$ is given by $(36 \times 8 + 59 \times 25)/25$. However, since Foldcomp is designed to encode all backbone atoms (C, N, $C\alpha$) instead of merely $C\alpha$, it is unfair to compare in this way. We thus also report its performance on all backbone atoms for reference.

### C.3 ABLATION STUDY SETTINGS

In this section, we describe the details settings for ablation studies in Figures 5b and 5c.

**Experiments without Linear Reparameterization:** We simply set $\mathbf{w} = \mathbf{h_w}$ without the linear matrix $\mathbf{A}$. Besides, since in this case, $\mathbf{w}$ follows mean-field Gaussian, we use the local reparameterization trick with 1 sample to reduce the variance during both training and inferring.

**Experiments without Positional Encodings:** Recall that the inputs of INRs in RECOMBINER is the concatenation of Fourier transformed coordinates $\gamma(\mathbf{x}_i)$ and the upsampled positional encodings at the corresponding position $\mathbf{z}_i = \phi(\mathbf{h_z})_{\mathbf{x}_i}$. In the experiments without positional encodings, we only input the Fourier transformed coordinates to the INR. To keep the INR size consistent, we also increase the dimension of the Fourier transformation, so that $\dim(\gamma'(\mathbf{x}_i)) \leftarrow \dim(\gamma(\mathbf{x}_i)) + \dim(\mathbf{z}_i)$. Also, we no longer need to train the upsampling network $\phi$.

**Experiments without Hierarchical Model:** We assume all patch-INRs are independent and simply assign independent mean-field Gaussian priors and posteriors over $\mathbf{h_w}^{(\pi)}$ for each patch.

**Experiments without Random Permutation across patches:** Recall in RECOMBINER, for each level in the hierarchical model, we stack the representations together into a matrix, where each row is one representation. We then (a) apply the same permutation over all rows. This is the same as COMBINER and is to ensure KL is distributed uniformly across the entire representation for each patch. Then (b) for each column, we apply its own permutation to encourage KL to be distributed uniformly across patches. In the ablation study, we do not only apply the permutation in (b) but still perform the permutation in (a).

# D SUPPLEMENTARY EXPERIMENTS AND RESULTS

## D.1 METHODS VISUALIZATION

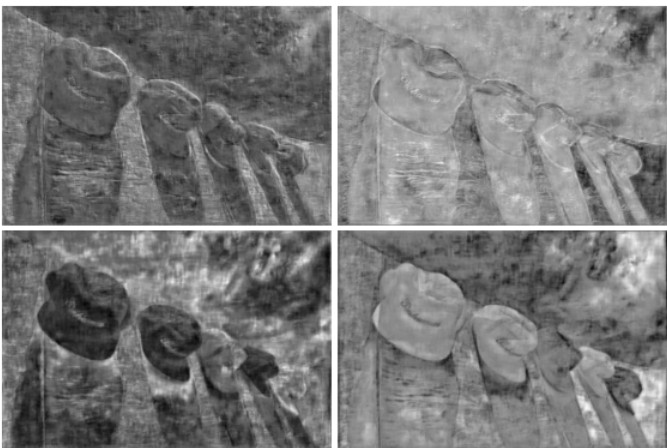

(a) Visualization of 4 channels in the upsampled positional encodings for `kodim03` at 0.488 bpp. Patches are stitched together for a clearer visualization.

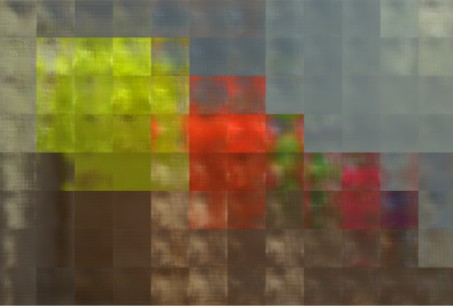

(b) Visualization of the information contained in encoded $\mathbf{h_w}$ for `kodim03` at 0.488 bpp. Patches are stitched together.

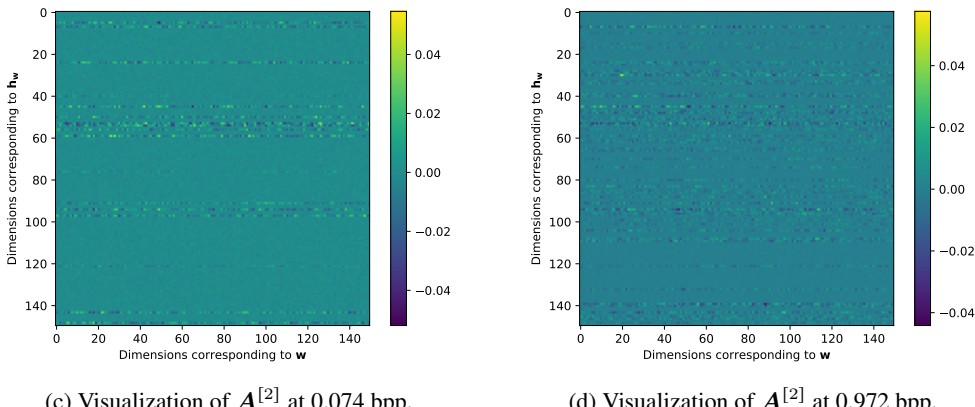

(c) Visualization of $\boldsymbol{A}^{[2]}$ at 0.074 bpp.       (d) Visualization of $\boldsymbol{A}^{[2]}$ at 0.972 bpp.

Figure 7: Visualizations.

In this section, we bring insights into our methods by visualizations. Recall that each signal is represented by $\mathbf{h_Z}$ and $\mathbf{h_w}$ together in RECOMBINER. We visualize the information contained in each of them. Besides, we visualize the linear transform $\boldsymbol{A}$ to understand how it improves performances.

**Positional encodings**: We take `kodim03` at 0.488 bpp as an example, and visualize 4 channels of its *upsampled* positional encodings $\phi(\mathbf{h_z})$ in Fig 7a. Interestingly, before fed into the INR, the positional

encodings already present a pattern of the image. This is an indication of how the learnable positional encodings help with the fitting. When the target signal is intricate, and there is a strict bitrate constraint, the INR capacity is insufficient for learning the complex mapping from coordinates to signal values directly. On the other hand, when combined with positional encodings, INR simply needs to extract, combine, and enhance this information, instead of "creating" information from scratch. This aligns with the findings of the ablation study, which indicate that learnable positional encodings have a more significant impact on CIFAR-10 at low bitrates and the Kodak dataset, but a small effect on CIFAR-10 at high bitrates.

**Information contained in $\mathbf{h_w}$**: To visualize the information contained in $\mathbf{h_w}$, we also take `kodim03` at 0.488 bpp as an example. We reconstruct the image using $\mathbf{h_w}$ for this image but mask out $\mathbf{h_Z}$ by the prior mean. The image reconstructed in this way is shown in Fig 7b.

From the figure, we can clearly see $\mathbf{h_w}$ mostly captures the color specific to each patch, in comparison to the positional encodings containing information more about edges and shapes. Moreover, interestingly, we can see patches close to each other share similar patterns, indicating the redundancy between patches. This explains why employing the hierarchical model provides substantial gains, especially when applying it together with positional encodings.

**Linear Transform $A$**: To interpret how the linear reparameterization works, we take the Kodak dataset as an example, and visualize $A$ for the second layer (i.e., $A^{[2]}$) at 0.074 and 0.972 bpp in Fig 7c and 7d. Note that this layer has 32 hidden units and thus $A^{[2]}$ has a shape of $1056 \times 1056$. We only take a subset of $150 \times 150$ in order to have a clearer visualization. Recall $\mathbf{w} = \mathbf{h_w} A$, and thus rows correspond to dimensions in $\mathbf{h_w}$ and columns correspond to dimensions in $\mathbf{w}$.

It can be seen that when the bitrate is high, many rows in $A$ are active, enabling a flexible model. Conversely, at lower bitrates, many rows become 0, effectively pruning out corresponding dimensions. This explains clearly how $A$ contributes to improve the performance: first, $A$ greatly promotes parameter sharing. For instance, at low bitrates, merely 10 percent of the parameters get involved in constructing the entire network. Second, the pruning in $\mathbf{h_w}$ is more efficient than that in $\mathbf{w}$ directly. The predecessor of RECOMBINER, i.e., COMBINER, utilizes standard Bayesian neural networks. It controls its bitrates by pruning or activating the hidden units. When a unit is pruned, the entire column in the weight matrix will be pruned out (Trippe & Turner, 2017). In other words, in COMBINER, the pruning in $\mathbf{w}$ is always conducted *in chunks*, which highly limits the flexibility of the network. On the contrary, in our approach, the linear reparameterization enables a direct pruning or activating of each dimension in $\mathbf{h_w}$ individually, ensuring the flexibility of INR while effectively managing the rate.

Another interesting observation is the matrix $A$ essentially learns a low-rank pattern without manual tuning. This is in comparison with VC-INR (Schwarz et al., 2023) where the low-rank pattern is explicitly enforced by manually setting the LoRA-style (Hu et al., 2021) modulation.

## D.2 EFFECTIVENESS OF RANDOM PERMUTATION

In this section, we provide an example illustrating the effectiveness of random permutation across patches. Specifically, we encode `kodim23` at 0.074 bpp, both with and without random permutation, and visualize their residual images in Figure 8. We can see that, without permutation, the residuals for complex patches are significantly larger than simpler patches. This is due to the fact that, in RECOMBINER, the bits allocated to each patch are merely determined by the number of blocks, which is shared across all the patches. On the other hand, after the permutation, we can see a more balanced distribution of residuals across patches: complex patches achieve better reconstructions, whereas simple patches' performances only degrade marginally. This is because, after the permutation across patches, each block can have different patches' representations, enabling an adaptive allocation of bits across patches. Overall, random permutation yields a 1.00 dB gain on this image.

## D.3 INFLUENCE OF SAMPLE SIZE

As discussed in Appendix C.1, in our experiments, we use 5 samples to estimate the expectation in the $\beta$-ELBO in Equation (1), when inferring the posterior of a test datum. Here, we provide the RD

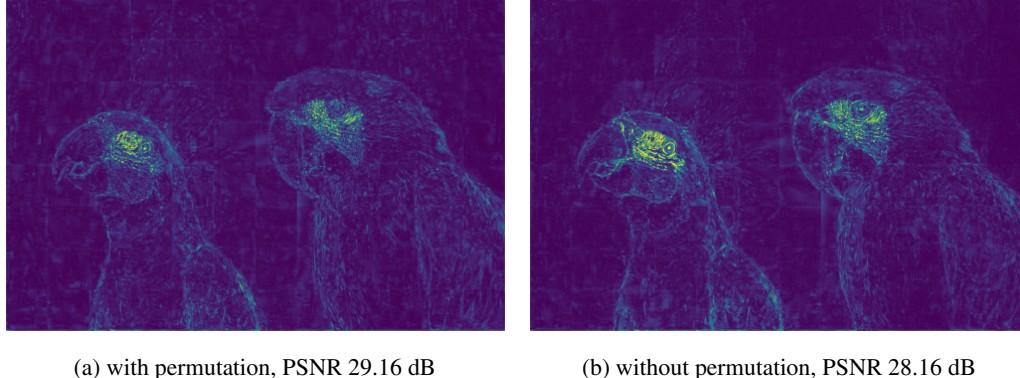

(a) with permutation, PSNR 29.16 dB          (b) without permutation, PSNR 28.16 dB

Figure 8: Comparison of residuals of `kodim23` at 0.074 bpp, with and without random permutation across patches.

curve using 1, 5 and 10 samples, on 500 randomly selected Cifar-10 test images and `kodim03` as examples, to illustrate the influence of different choices of sample sizes.

As shown in Figure 9, the sample size mainly impacts the performance at high bitrates. Besides, further increasing the sample size to 10 only brings a minor improvement. Therefore, we choose 5 samples in our experiments to balance between encoding time and performance. It is also worth noting that using just 1 sample does not significantly reduce the performance. Therefore, we have the flexibility of choosing smaller sample sizes when prioritizing encoding time, with minor performance impacts.

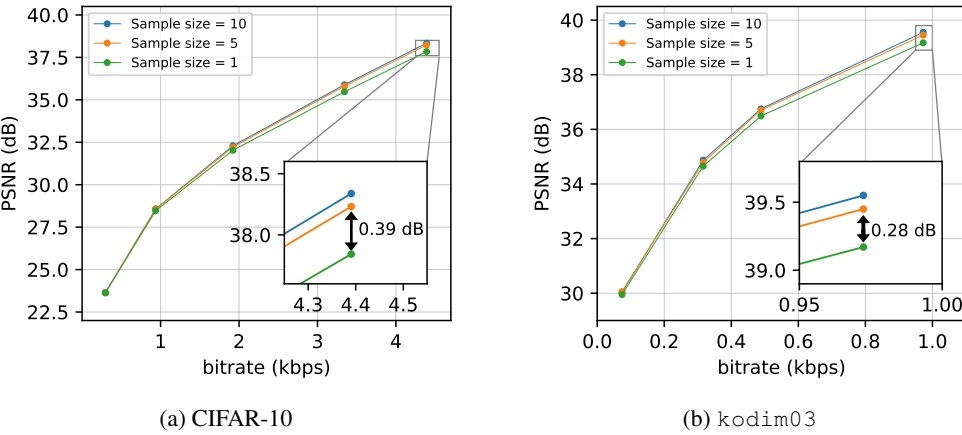

(a) CIFAR-10                         (b) `kodim03`

Figure 9: Influence of Sample size. (a) RD curve evaluated on 500 randomly selected CIFAR-10 images. (b) RD curve evaluated on `kodim03`.

### D.4 ROBUSTNESS DURING TRAINING

Different from previous INR-based codecs based on MAML (Finn et al., 2017) including COIN++ (Dupont et al., 2022), MSCN (Schwarz & Teh, 2022) and VC-INR (Schwarz et al., 2023), our proposed RECOMBINER does not require nested gradient descent and thus features higher stability during training period.

To demonstrate this advantage, we present a visualization of the average $\beta$-ELBO during training on CIFAR-10 across three bitrates in Figure 10. We can see that the training curves exhibit an initial dip followed by a consistent increase. The dip at the beginning is a result of our adjustment of $\beta$ during training (Step 3 in Algorithm 1). Importantly, this adjustment does not impact training robustness; and we can see that $\beta$ is quickly adjusted, and the training proceeds smoothly.

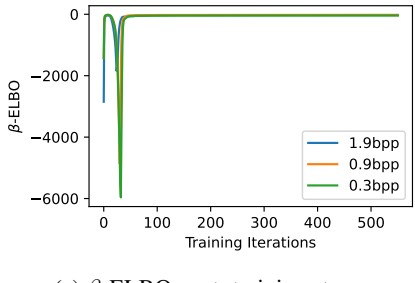

(a) $\beta$-ELBO w.r.t. training steps.

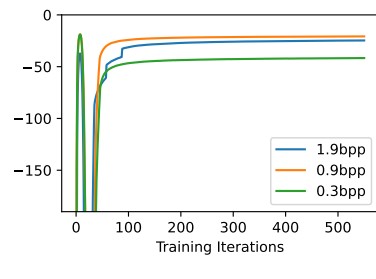

(b) Zoom-in plot.

Figure 10: Average training $\beta$-ELBO on Cifar-10 at three different bitrates. The initial dip is because we also adjust $\beta$ during training to ensure the coding budget (Step 3 in Algorithm 1). We can see the initial $\beta$ quickly adjusts in the first several steps, and then the training proceeds smoothly.

### D.5 CODING TIME

In this section, we provide details regarding the encoding and decoding time of RECOMBINER. The encoding speed is measured on a single NVIDIA A100-SXM-80GB GPU. On CIFAR-10 and protein structures, we compress signals in batch, with a batch size of 500 images and 1,000 structures, respectively. On Kodak, audio, and video datasets, we compress each signal separately. We should note that the batch size does not influence the results. Posteriors of signals within one batch are optimized in parallel, and their gradients are not crossed. The decoding speed is measured per signal on CPU.

Similar to COMBINER, our approach features a high encoding time complexity. However, the decoding process is remarkably fast, even on CPU, matching the speed of COIN and COMBINER. Note that the decoding time listed here encompasses the retrieval of samples for each block. In practical applications, this process can be implemented and parallelized using lower-level languages such as C++ or C, which can lead to further acceleration of execution.

| Bitrate | Encoding Time (GPU, 500 instances) | Decoding Time (CPU, per instance) |
|---|---|---|
| 0.297 bpp | ∼63 min | 0.00386 s |
| 0.719 bpp | ∼65 min | 0.00429 s |
| 0.938 bpp | ∼68 min | 0.00461 s |
| 1.531 bpp | ∼72 min | 0.00514 s |
| 1.922 bpp | ∼75 min | 0.00581 s |
| 3.344 bpp | ∼87 min | 0.00776 s |
| 4.391 bpp | ∼93 min | 0.01050 s |

Table 3: Coding time for CIFAR-10.

| Bitrate | Encoding Time (GPU, per instance, 96 patches) | Decoding Time (CPU, per instance) |
|---|---|---|
| 0.074 bpp | ∼59 min | 0.25848 s |
| 0.130 bpp | ∼64 min | 0.29117 s |
| 0.178 bpp | ∼67 min | 0.30875 s |
| 0.316 bpp | ∼72 min | 0.29690 s |
| 0.488 bpp | ∼80 min | 0.34237 s |
| 0.972 bpp | ∼92 min | 0.41861 s |

Table 4: Coding time for Kodak.

| Bitrate | Encoding Time (GPU, per instance, 50 patches) | Decoding Time (CPU, per instance) |
|---|---|---|
| 5.69 kbps | ~18 min | 0.05564 s |
| 10.66 kbps | ~21 min | 0.06003 s |
| 22.11 kbps | ~22 min | 0.06166 s |
| 43.64 kbps | ~22 min | 0.07350 s |

Table 5: Coding time for Audio.

| Bitrate | Encoding Time (GPU, per instance, 64 patches) | Decoding Time (CPU, per instance) |
|---|---|---|
| 0.115 bpp | ~49 min | 0.31936 s |
| 0.244 bpp | ~62 min | 0.33416 s |
| 0.605 bpp | ~78 min | 0.33448 s |
| 1.183 bpp | ~102 min | 0.35665 s |

Table 6: Coding time for Video.

| Bitrate | Encoding Time (GPU, 1000 instance) | Decoding Time (CPU, per instance) |
|---|---|---|
| 11.17 bpa | ~72 min | 0.00704 s |
| 35.17 bpa | ~123 min | 0.00948 s |
| 60.67 bpa | ~175 min | 0.01429 s |
| 83.83 bpa | ~226 min | 0.01778 s |
| 106.17 bpa | ~274 min | 0.02014 s |

Table 7: Coding time for Protein.

# E  THINGS WE TRIED THAT DID NOT WORK

- in RECOMBINER, we apply linear reparameterization on INR weights, which transfers the weights linearly into a transformed space. Perhaps a natural extension is to apply more complex transformations, e.g., neural networks, or flows. We experimented with this idea, but it did not provide gains over the linear transformation.

- in RECOMBINER, we propose a hierarchical Bayesian model, equivalent to assigning hierarchical hyper-priors and inferring the hierarchical posteriors over the means of the INR weights. A natural extension can be assigning hyper-priors/posteriors to both means and variances. But we did not find any gain by this.

- in RECOMBINER, the hierarchical Bayesian model is only applied to the latent INR weights $h_w$. It is natural to apply the same hierarchical structure to the latent positional encodings $h_z$. However, we found it does not provide visible gain.

# F  MORE RD CURVES

Here, we show the full-resolution RD curves for image compression in Figures 11 and 12. Besides, we also provide a further comparison between RECOMBINER with COMBINER on 24 test audio clips from LibriSpeech in Figure 13.

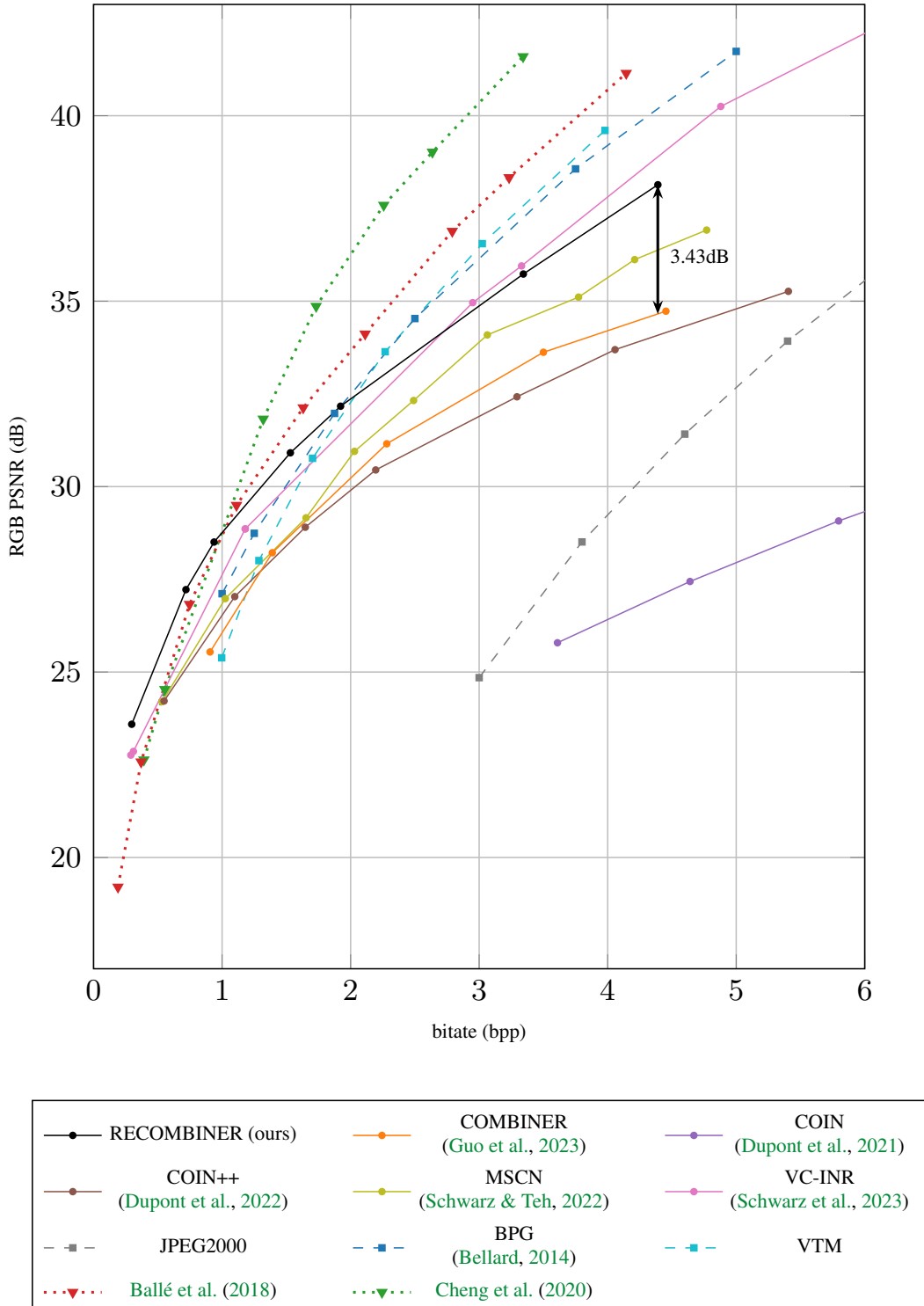

Figure 11: RD curve on CIFAR-10.

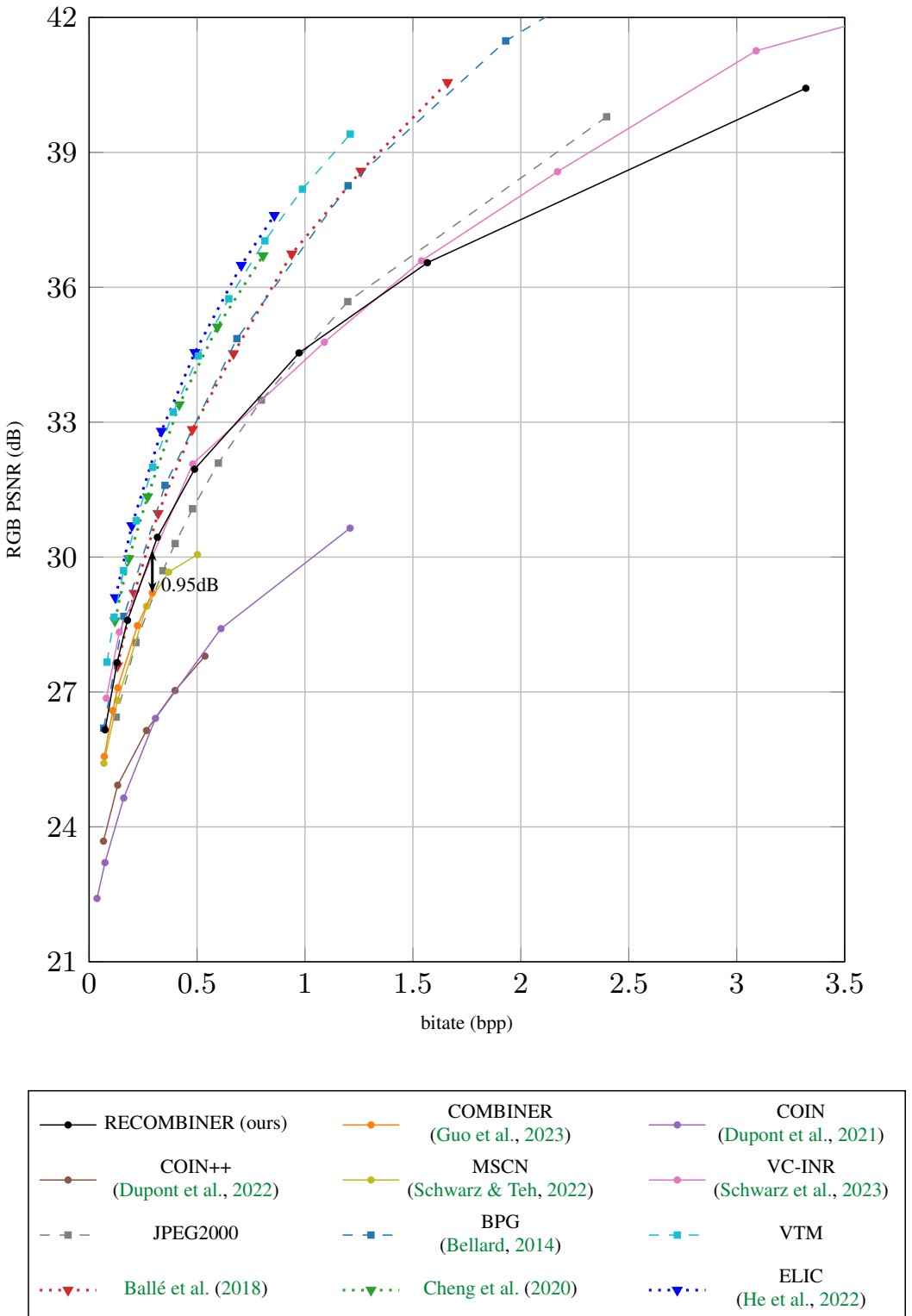

Figure 12: RD curve on Kodak.

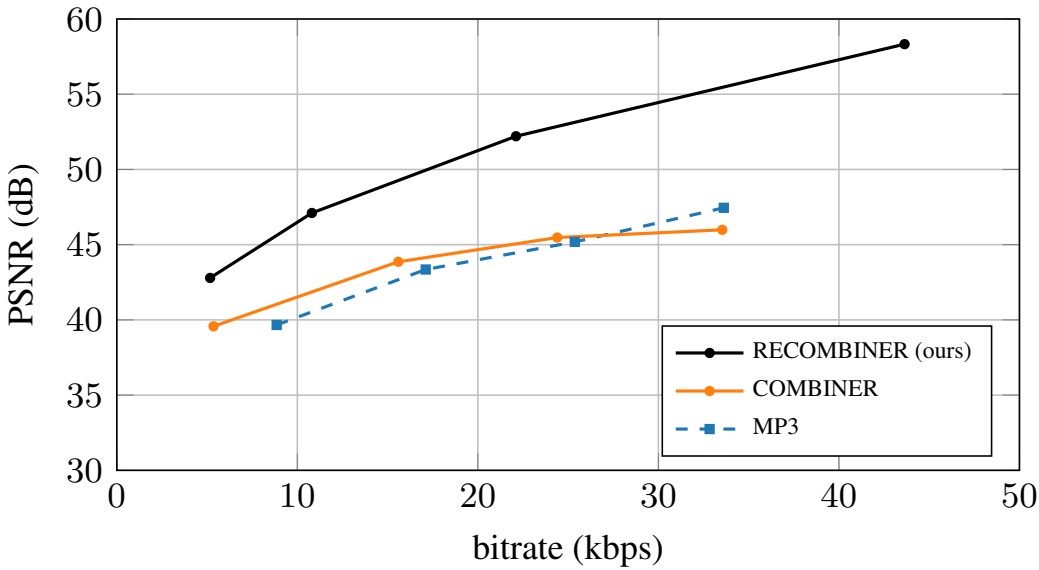

Figure 13: RD curve of MP3, COMBINER and RECOMBINER on 24 test audio clips from LibriSpeech test set.

## G  RD VALUES

**CIFAR-10:**

```
rate = [0.297, 0.719, 0.938, 1.531, 1.922, 3.344, 4.391]
```
```
PSNR = [23.592, 27.222, 28.505, 30.911, 32.168, 35.732, 38.139]
```

**Kodak:**

```
rate = [0.074, 0.130, 0.178, 0.316, 0.488, 0.972, 1.567, 3.320]
```
```
PSNR = [26.158, 27.653, 28.594, 30.439, 31.953, 34.540, 36.547,
40.426]
```

**Audio:**

On full test set:

```
rate = [5.685, 10.661, 22.112, 43.637]
```
```
PSNR = [42.612, 47.101, 52.196, 58.195]
```

On 24 test examples (to compare with COMBINER):

```
rate = [5.168, 10.805, 22.112, 43.637]
```
```
PSNR = [42.789, 47.106, 52.206, 58.327]
```

**Video:**

```
rate = [0.115, 0.244, 0.605, 1.183]
```
```
PSNR = [28.722, 31.494, 35.717, 39.171]
```

**Protein:**

```
rate = [11.17, 35.17, 60.67, 83.83, 106.17]
```
```
RMSD = [0.9242, 0.1388, 0.0709, 0.0506, 0.0436]
```

## H  MORE DECODED EXAMPLES

### H.1  CIFAR-10

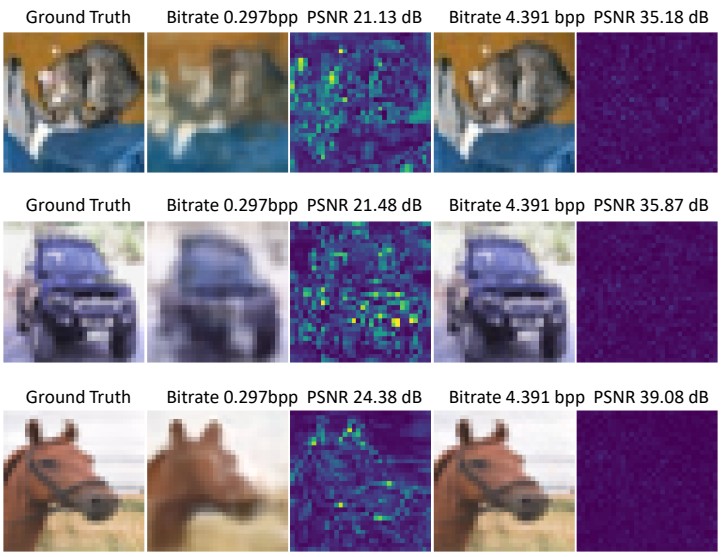

Figure 14: Decoded CIFAR-10 images and residuals.

### H.2  KODAK

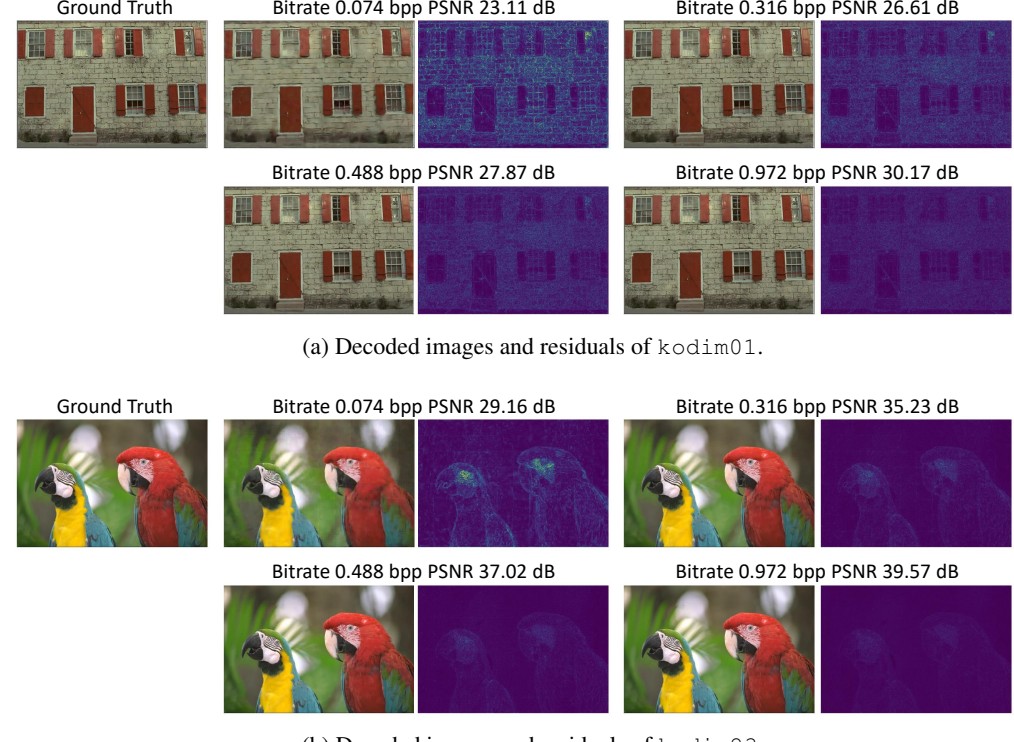

(a) Decoded images and residuals of `kodim01`.

(b) Decoded images and residuals of `kodim23`.

Figure 15: Examples of decoded Kodak images and their residuals.

## H.3 AUDIO

| Decoded Audios | | | Ground Truth |
|---|---|---|---|
| 5.17 kbps, 46.78 dB | 10.81 kbps, 51.53 dB | 22.11 kbps, 56.45 dB | |
| here | here | here | here |

Table 8: Decoded audio examples.

## H.4 VIDEO

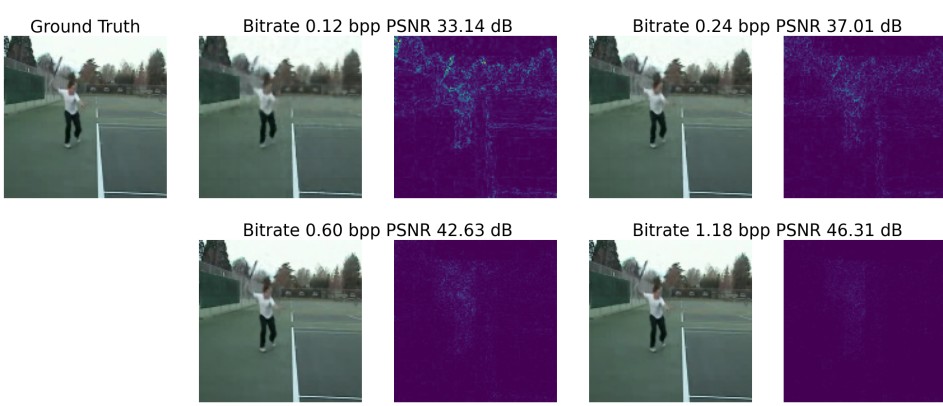

Figure 16: Examples of decoded videos and residuals. Animation visualization is available here.

## H.5 PROTEIN STRUCTURE

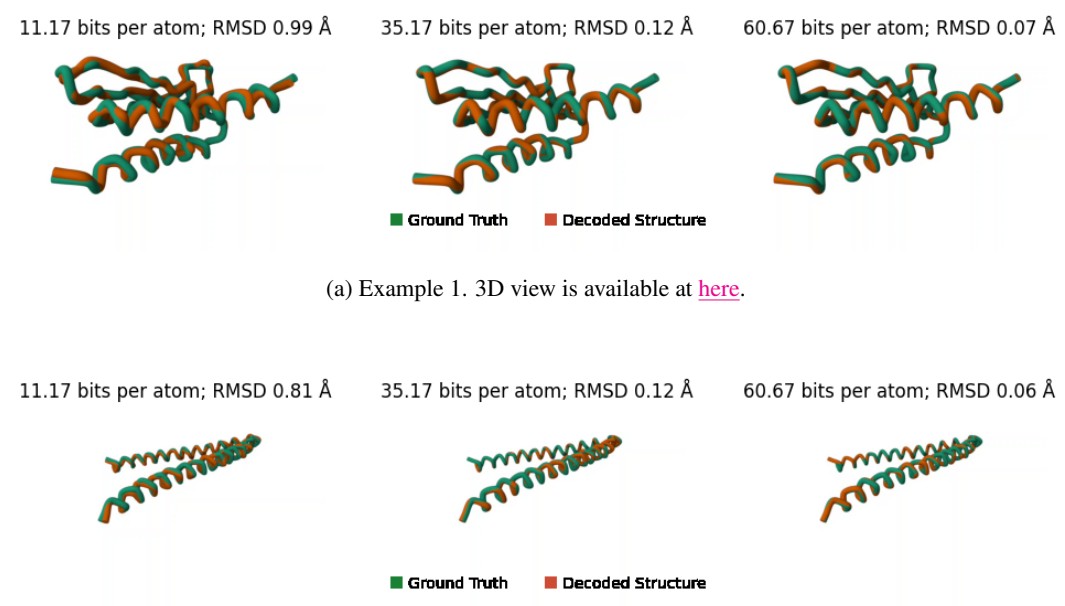

(a) Example 1. 3D view is available at here.

(b) Example 2. 3D view is available at here.

Figure 17: Examples of decoded protein structures and their ground truths.

