# OpenReview forum: "RECOMBINER: Robust and Enhanced Compression with Bayesian Implicit Neural Representations"
_ICLR.cc/2024/Conference — ICLR 2024 poster_

### Official Review · Reviewer_DDkv · 2023-10-31

**Soundness:** 3 good
**Presentation:** 3 good
**Contribution:** 3 good
**Rating:** 6
**Confidence:** 3

**Summary:**

The paper introduces RECOMBINER, an extension of the previous COMBINER implicit neural representation method for achieving neural compression. The new work includes a few model variations from the original COMBINER, including 1) a model parameter reparametrization, 2) the inclusion of positional encodings, and 3) a patch processing mechanism to handle large images. For these new model complexities, the paper also proposes an altered training procedure for fitting the parameters and the variational model. The paper considers numerical experiments for image, audio, video, and protein data. On image data, the proposed method outperforms VAE-based alternatives at low bitrates. On audio data, the proposed method outperforms the compared methods. On video data, the proposed method outperforms H.264/H.265 when H.264/H.265 are not in quality mode. The paper also validates its changes with ablation experiments.

**Strengths:**

1. The paper proposes a series of changes in COMBINER that result in good improvements to performance on all tasks where there is data for both methods (RECOMBINER and COMBINER were not compared for video and protein data).
2. The paper looks a large variety of tasks - most compression papers would only focus on one of these tasks.
3. The mathematics for the reparametrization are presented clearly and intuitively.
4. The patch-level processing for high-resolution data is a particularly welcome modification for image compression.

**Weaknesses:**

I am waffling on this paper a little bit largely due to the comparisons. Some of the tasks seem a little bit selective in terms of baselines. I raise a few points below.

1. Despite the improvements, overall INRs continue lag behind performance of competing methods on 3 out of 4 tasks.
2. INR performance comes at a compute cost penalty. This would be particularly large in the case of the video and image codec comparisons.
3. Errors for the protein data seem quite large, and I don't think the benefit of rate control is particularly useful for this setting. The paper presents RECOMBINER as an option here, but when the rate is lower than the competing methods, the error approaches the machine resolution. RECOMBINER only matches competitor compression performance at high rates.
4. Many neural methods for compression (particularly for audio) rely on perceptual compression rather than rate-distortion as is experimented with in the present paper. The paper does not consider how RECOMBINER could be adapted to become a perceptual codec.
5. Older image compression methods are used as baselines. The handcrafted baseline of choice is now VTM, and the baseline neural image compression is ELIC (He, 2022).

He, Dailan, et al. "Elic: Efficient learned image compression with unevenly grouped space-channel contextual adaptive coding." Proceedings of the IEEE/CVF Conference on Computer Vision and Pattern Recognition. 2022.

**Questions:**

1. Do you think there is a mechanism where RECOMBINER could be adapted to a perceptual codec?
2. Did you consider other neural audio compression methods? Encodec (Defossez, 2022) is one of particular note.
3. How are Fourier embeddings computed for protein and video data?

Défossez, Alexandre, et al. "High fidelity neural audio compression." arXiv preprint arXiv:2210.13438 (2022).

---

> ### Author Response · Authors · 2023-11-17
> **Response to Reviewer DDkv**
>
> We thank the reviewer for their review and suggestions to improve our paper. Below, we respond to the reviewer’s concerns. We are happy to discuss any further concerns the reviewer might have. However, should we have addressed the reviewer’s concerns and questions, we kindly invite them to raise their score.
>
> > Some of the tasks seem a little bit selective in terms of baselines.
>
> We agree with the reviewer and have updated our paper with modifications:
>  1) We have updated our paper to include VC-INR in our audio compression plot in Figure 3b and plan to include it for video compression in Figure 3c. Please see our response to Reviewer oN2Z for a detailed explanation of how we ensure a fair comparison between the methods.
>  2) We added ELIC and VTM as baselines in Figure 3a. We find that they do not outperform RECOMBINER on CIFAR-10 at low bitrates, and our method is still SOTA in this region. Interestingly, VTM performs even worse than BPG on Cifar-10 at lower bitrates. This might be because VVC is designed for higher resolution. Thus, advanced techniques such as block partitioning and mode selection in VVC might be less effective when compressing low-resolution images. We also tested ELIC on CIFAR-10, but it did not work well on low-resolution images. Its performance was even worse than Balle 2018 and sometimes does not converge. Therefore, we omit the ELIC curve on CIFAR-10.
>
> **INR performance gap and slow encoding:** While these are undoubtedly important observations, we argue that they are general limitations of INR-based approaches and aren’t weaknesses of RECOMBINER per se. We also note that VAE-based methods are much more mature than INRs and rely on specifically designed network architecture and powerful entropy models. Moreover, INRs offer unique advantages compared to VAE-based methods, such as data modality-agnosticism, lightweightness, and fast decoding times on even CPUs.
>
> **Errors in protein data:** As shown at https://www.rcsb.org/stats/distribution-resolution, most structures have a resolution around 2Å, and only a very small minority have a resolution below 1Å. Thus, our errors are still much less than the machine resolution; hence, we believe they should be insignificant for any downstream task.
>
> Furthermore, unlike the baselines, our method is highly extensible. For example, it can be easily extended to capture the rotation/translation-invariant by modifying the loss function. It can also take the protein’s uncertainty into account with ease. As another example, it is known that the alpha-helix region of the protein has lower uncertainty than coils. We can capture this information by appropriately weighting the loss function, which ensures that the compression budget is allocated to the more important regions, such as the alpha-helix. In our paper, our main goal for this experiment is to demonstrate RECOMBINER’s modality-agnosticism and leave these specific extensions for future work.
>
> **Perceptual compression:** We agree that this is a promising direction and are currently working on incorporating perceptual constraints into INR-based compression. However, we emphasize that all current INR-based compression methods focus on MSE rate-distortion performance. Therefore, we do not believe that this is one of our weaknesses.
>
> **Regarding the Reviewer’s questions:**
>  1. Yes, one way to adapt RECOMBINER to a perceptual codec is to add an adversarial term in the loss function. This term can come from an extra discriminator.
>  2. As discussed above, no INR-based method is currently designed for perceptual loss. But we will be happy to consider and compare with the work you mentioned once we successfully adapt RECOMBINER to a perceptual codec. We will also update our discussion section in our camera-ready version to note this future direction.
>  3. For a $M$-dimensional modality, we can write the coordinate as $\mathbf{x} = [x_1, ..., x_M]^\top$. We define its Fourier embeddings $ \gamma(\mathbf{x}) \in \mathbb{R}^D$ as
> \begin{align}
>    \gamma(\mathbf{x}) = \Big[\cos(  a^{{0}/{d}} \pi x_1 ), &\dots, \cos(  a^{{0}/{d}} \pi x_M ), \\\\
>    \cos(  a^{{1}/{d}} \pi x_1 ), &\dots, \cos(  a^{{1}/{d}} \pi x_M ), \\\\
>    &\vdots\\\\
>     \cos(  a^{{d}/{d}} \pi x_1 ), &\dots, \cos(  a^{{d}/{d}} \pi x_M ), \\\\
>     \sin(  a^{{0}/{d}} \pi x_1 ), &\dots, \sin(  a^{{0}/{d}} \pi x_M ), \\\\
>    \sin(  a^{{1}/{d}} \pi x_1 ), &\dots, \sin(  a^{{1}/{d}} \pi x_M ), \\\\
> &\vdots\\\\
>     \sin(  a^{{d}/{d}} \pi x_1 ), &\dots, \sin(  a^{{d}/{d}} \pi x_M )\Big]^\top\text{,}
> \end{align}
> For image, $M=2$; for audio and protein, $M=1$ (note that the backbone atoms of protein are located in a chain); for video, $M=3$.
> In our experiments, we follow COMBINER, setting $d = \lfloor\frac{D}{2M}\rfloor-1$ and $a=1024$.

---

> > ### Comment · Reviewer_DDkv · 2023-11-20
> >
> > I would like to thank the authors for responding to my comments. I have increased my score for the paper.

---

### Official Review · Reviewer_TTUk · 2023-11-03

**Soundness:** 3 good
**Presentation:** 2 fair
**Contribution:** 3 good
**Rating:** 6
**Confidence:** 4

**Summary:**

This paper improves the most recent implicit neural representation (INR) compression method, called the COMBINER, and proposes a robust and enhanced COMBINER, thus named as RECOMBINER. The enhancements are mainly from the layer-wise block diagonal matrix in factorised Gaussian assumptions to increase the flexibility of overfitting data, additional positional embedding to address the local patterns, as well as the hierarchical model to compress high-resolution images. Experimental results verify the effectiveness of this improved COMBINER method.

**Strengths:**

1. This paper improves the existing COMBINER method, with robust and enhanced performances on data compression. The improvements are reasonably established, upon the stringent factorised Gaussian assumption as proposed in the original COMBINER method. A block-wise diagonal method does much help during training and inferencing.
2. This paper proposes the hierarchical strategy to accommodate the compression for high-resolution images. The optimisation is supported by minimising the upper bound when splitting into patches.
3. Experimental results have verified the effectiveness of the proposed RECOMBINER method. Although not beating the state-of-the-art VAE based methods, I still value this work to be a promising alternative direction for learnt data compression.

**Weaknesses:**

1. The quality of this paper needs to be comprehensively improved, whereby many typos exist. For example, "have a neural network memorize the data (Stanley, 2007) and encode the network weights instead.".
2. For the linear reparameterization module, why A^[l] is updated during the training stage? Also the authors claim that the block-wise diagonal matrix operates as good as the full covariance matrix. Is any verification on this, from either theoretically or emperically?
3. For the learnt positional embeddings, I am a bit confused on using additional position cues. Since x_i also includes the coordinate information, why using z_i helps to address the global representation challenge?

**Questions:**

Please see my weakness.

---

> ### Author Response · Authors · 2023-11-17
> **Response to Reviewer TTUk**
>
> We thank the reviewer for their constructive feedback on our paper. We are delighted that the reviewer appreciates our technical contributions and values our work. We respond to the reviewer’s concerns below. We are also happy to discuss any further questions the reviewer might have.
>
> **Typo:** We thank the reviewer for their feedback and have clarified the sentence they highlighted. We will also carefully re-read our manuscript and correct any typos.
>
> **Linear reparameterization:** The matrix $A$ controls the correlation between the network weights. This correlation will depend on the bitrate and the data modality and thus cannot be handcrafted in advance. Hence, we learn it from the training data. To better understand what matrix $A$ has learned, we provide the visualization of $A$ in Figure 7(c)(d) in the Appendix. By updating in the training period, we can see that $A$ is specifically tailored for different bitrates: some columns are pruned out (i.e., almost 0), and some are activated. Note that before the training, we randomly initialize the matrix $A$. So, this visualization illustrates what $A$ has learned by updating.
>
> We found that models trained with full covariance matrices performed very similarly to the ones using layer-wise block diagonal matrices in practice but were much more expensive to train/evaluate. We also note that the layer-wise block diagonal matrix is a standard approximation widely used for neural networks. For example, KFAC also assumes independence between layers [1].
>
> **Positional encoding:** At a high level, the $z_i$s just allow more flexibility for the model and can be thought of as “zeroth-level bias.” We visualize the learned positional encodings for a Kodak image in Figure 7a in the Appendix to illustrate their effect.
>
> ## References
>
> [1] Martens, J., & Grosse, R. (2015, June). Optimizing neural networks with Kronecker-factored approximate curvature. In International conference on machine learning (pp. 2408-2417). PMLR

---

> > ### Comment · Reviewer_TTUk · 2023-11-22
> >
> > Many thanks for the reviewers' response. I would like to keep my score by reading the response and other reviewers' comments.

---

### Official Review · Reviewer_oN2Z · 2023-11-04

**Soundness:** 4 excellent
**Presentation:** 4 excellent
**Contribution:** 2 fair
**Rating:** 8
**Confidence:** 4

**Summary:**

This paper proposes three additional techniques that can be used to improve COMBINER, an INR-based modality-agnostic data compression framework. (1) Linear Reparameterization: One places Gaussian factorized priors on the latent codes, instead of the weight parameters themselves. (2) Learnable Positional Encodings: One generates positional encoding from some lower-dimensional learnable latent code, which is mapped to the full-dimensional space through upsampling and convolution. (3) Hierarchical Bayesian modeling on patchified data: One divides the data into patches and models it with hierarchical Bayesian approach. These changes make the modified version (called RECOMBINER) achieve the performance comparable to SOTA codecs and a competing INR-based approach (VC-INR).

**Strengths:**

- The performance gain over the previous attempt in this direction (i.e., COMBINER) is indeed very impressive. The innovations introduced in this paper seems to make the combiner-like approach a competitive paradigm for INR-based data compression.
- All three proposed modifications are technically well-designed and well-motivated. Especially, the hierarchical Bayes approach in section 3.3 looks quite clear and intuitive.
- The writing is very clear. The paper is one of the most easy-to-read papers among all papers I read in the past several months.
- The empirical validation is quite extensive, covering image/video/audio to protein structures.
- Appendix E, which describes the things that didn't work, is very useful and a good academic practice.

**Weaknesses:**

- **R-D tradeoff on image.** While it is definitely good to see that recombiner outperforms combiner, the performance does not clearly outperform VC-INR, an even older baseline. I do not see this as a very big drawback, but this observation makes it quite questionable whether combiner-like approach has any great potential in the long run.

- **Comparison on audio/video.** I wonder how the RECOMBINER compare with VC-INR and COIN++ on audio/video datasets. The paper says that it does not compare with these baselines, because "the works use different data splits." I do not think this is a good excuse to not compare with these baselines. It seems evident that recombiner outperforms combiner, but in the end, we would like to understand whether the combiner-like approach is indeed a useful framework, when compared with other INR-based paradigms (and furthermore, VAE-based ones).

- **Limited range of bitrate in comparison---problems in extending to higher bitrates?.** Comparing with baselines, the range of bitrates considered is considerably smaller. For Kodak, VC-INR compares on the bitrate up to 3.5, while this work only considers up to 1.2. For videos, the range is again up to 1.2, while VC-INR does it to over 4. The same for the audio. I wonder why the authors made this choice. Does this mean that recombiner has difficulties in training in high bpp regime?

- **(minor) Practicality.** Not a big issue for a research-in-progress, but the long encoding/decoding time is a severe practical limitation (appendix d.4). I wonder how these computational costs compare with the baselines; it would be a great help if authors could give an explicit head-to-head comparison with combiner, coin++, and vc-inr.

- **(minor) Limited Impact of Technical Contributions.** Given the lukewarm performance gain over competing paradigms, it would be great if the proposed technical innovations are very novel or have a wider applicability to other fields of machine learning. I am not sure if this is the case; the linear reparameterization and learned positional encodings are either not very new or highly specialized to the context of recombiner. I do appreciate the technicality of the hierarchical Bayes part, but some components are somewhat mysterious to me (the strange importance of random permutation) and the idea may not have a wider impact outside this specific context.

- (minor) the last row of the legend in figure 5c is wrongly ordered?

**Questions:**

In addition to the requests in "weaknesses," I have one more question:
- The figure 5c is very interesting, in the sense that using the tricks without random permutation is almost the worst among all choices (brown). Do you have any explanation?

---

> ### Author Response · Authors · 2023-11-17
> **Response to Reviewer oN2Z, Part 1/3**
>
> We thank the reviewer for their detailed and insightful review. We are delighted that the reviewer found our paper well-written and recognized our method outperforms COMBINER and that we performed extensive empirical evaluation. We respond to the reviewer's concerns below. Should the reviewer find our answers satisfactory, we kindly invite them to consider raising their score.
>
> **R-D tradeoff on image.**
> We appreciate that RECOMBINER’s potential compared to other methods is not clear at first sight from the figures. However, we believe that RECOMBINER is more promising than previous INR-based methods because it does the "right thing" and jointly optimizes the rate-distortion tradeoff. Non-variational INR compression methods cannot do this. Even VC-INR, the most advanced technique, follows a two-stage training pattern:
>  1) Over a training set, it fits an INR to each datum using only the distortion as the objective. Thus, it obtains a new dataset of INR weights.
>  2) Then, it fits a variational autoencoder to the INR weight dataset inspired by Balle et al.'s model.
>
> Regarding VC-INR in particular, we would like to highlight three aspects of the work and contrast them with RECOMBINER:
>  1. From a theoretical perspective, VC-INR's performance is capped by the abovementioned first stage of training. We don't necessarily believe that this cap is a big problem in practice, but it's still worth noting that RE/COMBINER does not have this limitation.
>  2. We can view VC-INR's second training stage as building a complex entropy model for the weights, essentially equivalent to RECOMBINER's weight prior. Thus, considering that VC-INR's entropy model is given by a VAE while RECOMBINER's prior is multivariate Gaussian, it is quite encouraging that the two methods' performances match closely. Furthermore, this indicates that designing more elaborate weight priors for RECOMBINER (e.g., using normalizing flows) could significantly boost its performance.
>  3. As a more subtle point, in private communication with the authors of VC-INR, the authors highlighted to us that to obtain each point on VC-INR's rate-distortion curve, they trained several different INRs with varying sizes but equal rate constraints and always compressed the best-performing INR. They noted that this selection process was especially important to ensure good performance in the low-to-medium bpp range. In contrast, in our paper, we fixed a single architecture for all our experiments, even across data modalities, and trained and compressed only a single model to obtain each point. We could likely get significantly improved curves if we followed a similar "grid search" over architectures at each rate point.
>
> To further add to the future potential of the COMBINER-like approach, we also highlight that training RE/COMBINER is much more stable than training VC-INR, as it does not involve meta-learning. The authors of VC-INR note in their paper: “In line with prior work, we find the direct optimisation of non-linear networks via Meta-Learning to be unstable and under-performing. As low-rank parameterisations are also known to suffer from stability issues, the direct use yield unsatisfactory results.” Hence, they need to introduce several techniques to stabilize training. In contrast, RECOMBINER’s training is robust and stable without requiring stabilization techniques. To illustrate this, we added a plot of RECOMBINER’s training curve in Figure 10 in Appendix D.4. As the figure shows, there is a single dip in the training curve, which is due to the dynamic beta-adjustment procedure we describe in Section 3.4. After beta adjusts to its appropriate level early in the training, the training curves essentially monotonically increase without any dips.
>
> **Comparison on audio/video.**
>
> *Regarding COIN++:* We omitted COIN++ in the audio comparison, as on the reported bitrates, it performs worse than MP3, which RECOMBINER already outperforms significantly. Moreover, the authors of COIN++ do not report experiments on video data; hence, it is missing from that comparison plot.
>
> *Regarding VC-INR:* We agree with the reviewer and have changed our view. To ensure a fair comparison, we started evaluating RECOMBINER on the same test set on which VC-INR was evaluated. We will include the results in the camera-ready version of the paper. To get an idea of the performance, we included VC-INR in an indicative comparison plot in the updated manuscript on audio comparison in Figure 3b to include VC-INR. As we can see, the two methods have similar performance, with RECOMBINER even slightly outperforming VC-INR at a higher bitrate.
>
> Furthermore, we reached out to the authors of VC-INR to share the precise details of their train/test split on the video data so that we can provide a fair comparison to their method. We will add this comparison in the final version of the paper.

---

> ### Author Response · Authors · 2023-11-17
> **Response to Reviewer oN2Z, Part 2/3**
>
> **Limited range of bitrate in comparison---problems in extending to higher bitrates?**
> The reason for the limited range for the bitrate is quite prosaic: other than VC-INR, all other INR works focus on the smaller range highlighted by the reviewer. However, we emphasize that there are absolutely no issues with training RECOMBINER at higher bitrates. To demonstrate this point, we added Figures 11 and 12 in the appendix that compare the methods on an extended range on Kodak. We added a further high-bitrate point to the audio comparison in Figure 3b. As we can see, RECOMBINER remains competitive with VC-INR in this regime as well.
>
> **Practicality.**
> We agree this is a limitation of our work, as discussed in the limitation section. However, we highlight that our method’s decoding time is fast. Here, we provide the encoding time of COMBINER and RECOMBINER on Kodak for comparison. As we also mentioned in our paper, we can significantly reduce the coding time by reducing the sample size, with a minor impact on the performance. Therefore, we provide the reduced encoding time in this Table. Note that RECOMBINER is generally faster than COMBINER on Kodak due to our patching and parallelizing strategy.
>
> |               |       |                              |
> |:----------:|:----------------------------:|:----------------------------:|
> |               |   **RECOMBINER on Kodak**       |                              |
> |  Bitrates  |      Encoding Time (GPU)     |      Decoding Time (CPU)     |
> |  0.07 bpp  |            ~23 min           |           258.48 ms          |
> |  0.32 bpp  |            ~28 min           |           296.90 ms          |
> |        |                       **COMBINER on Kodak**        |                              |
> |  Bitrates  | Reported Encoding Time (GPU) | Reported Decoding Time (CPU) |
> |  0.07 bpp  |            ~21 min           |           348.42 ms          |
> |  0.29 bpp  |            ~79 min           |           602.32 ms          |
>
>
> As for COIN++, the author only reported the time complexity on CIFAR-10. Their reported time complexity is based on one image. While we follow COMBINER, compress 500 images in one batch. To fairly compare with them, we also report the average coding  time per image.  Since COIN++ is based on MAML, it only requires a handful of gradient descent steps during test time, and as such, its encoding complexity is lower them ours. The authors of VC-INR do not report runtime results, but since their method is also based on MAML, we expect their coding complexity to be lower than ours.
>
>
> |               |       |                              |
> |:----------:|:----------------------------:|:----------------------------:|
> |               |   **RECOMBINER on Cifar**       |                              |
> |  Bitrates  |      Encoding Time (*GPU*)    |      Decoding Time (*CPU*)     |
> |  0.94 bpp  |            ~3.6 s    (30 min / 500 images)        |           4.61 ms          |
> |  4.39 bpp  |            ~4.6 s  (38 min / 500 images)        |           10.50 ms          |
> |        |                       **COMBINER on Cifar**        |                              |
> |  Bitrates  | Reported Encoding Time (*GPU*) | Reported Decoding Time (*CPU*) |
> |  0.91 bpp  |            ~1.56 s  (13 min / 500 images)         |           2.06 ms          |
> |  4.45 bpp  |            ~4.08 s     (34 min / 500 images)        |           3.88 ms          |
> |  |                  **COIN++ on Cifar**            |                              |
> |            | Reported Encoding Time (*GPU*) | Reported Decoding Time (*GPU*) |
> |            |            94.9 ms           |            1.29 ms           |

---

> ### Author Response · Authors · 2023-11-22
> **Response to Reviewer oN2Z, Part 3/3**
>
> **Limited Impact of Technical Contributions.**
>
> *Linear reparameterization:* We agree that a linear reparameterization of the weights, in general, is not new. However, to the best of our knowledge,  the type of linear reparameterization we propose, i.e., using the same linear transform for the prior and the posterior, is novel in the context of Bayesian neural networks. It is particularly well-suited for any further work utilizing variational INRs, as the number of its variational parameters only grows linearly in the number of weights instead of quadratically (which would be the case if we learned a linear transform separately for each INR).
>
> *Random permutations:* We need the random weight permutation because we encode an INR’s weights sequentially, block-by-block. Ideally, when we infer the INR weight posterior, the KL divergence of the posterior from the prior in each block would conform to the bit budget we set (in the paper, we set a budget of 16 bits per block following COMBINER). However, this constraint is only enforced on average, and it is unlikely that each block will conform to the budget after inference. Thus, randomly permuting the weights allows us to distribute the KL better among the blocks.
>
> Regarding the specific random permutation technique we propose in Section 3.3 for patch-INRs: As we break an image up into patches, in addition to distributing the KL (information content) as evenly as possible, an additional aspect here is that we want to be able to encode the patch-INRs in parallel. With this in mind, the permutation technique we propose is essentially the simplest one that allows us to achieve these two goals simultaneously.
>
> **The figure 5c is very interesting, in the sense that using the tricks without random permutation is almost the worst among all choices (brown). Do you have any explanation?**
> As we mentioned in our previous point, randomly permutating the weights is crucial to ensure that block-wise relative entropy coding can work well. Especially regarding our patch-INRs, it ensures information content is spread as evenly as possible. Otherwise, we might spend too many bits on simple parts of the image (e.g., the sky), leaving insufficient bits for complex parts (e.g., one containing a person).
>
>
>
> *We are happy to answer any further questions the reviewer might have. If we answered the reviewer's questions adequately, we kindly invite the reviewer to consider raising their score.*

---

> ### Author Response · Authors · 2023-11-22
> **Thank you for your review! Please consider our response**
>
> We thank again the reviewer for their effort during the reviewing process. We believe we have addressed the stated concerns with our response, and we would like to ask the reviewer if they think this is the case as well. If the response is affirmative and based on the higher ratings provided by the other reviewers, we would like to ask the reviewer if they are happy to increase their score.

---

> > ### Comment · Reviewer_oN2Z · 2023-11-23
> > **Thank you for the response.**
> >
> > Dear authors,
> >
> > Thank you for the detailed response (with added experiments!), and sorry coming back too late. In general, I feel that most of my concerns have been addressed very well.
> >
> > One thing I would like to note is that, in my opinion, the added bpp range on Kodak (Fig 11) seems like it deserves some space in the main text. I won't necessarily agree to the authors' claim that "RECOMBINER remains competitive with VC-INR in this regime as well," but nevertheless it is much better to provide the full result.
> >
> > Regarding audio, I could not locate what * means---wish you be able to add them in the camera-ready.
> >
> > Regarding video, I sincerely wish that you add these eventually.
> >
> > Have raised the scores.
> >
> > Best,
> > Reviewer.

---

> ### Comment · Area_Chair_XUmL · 2023-11-23
> **Reviewer's feedback**
>
> Dear Reviewer oN2Z,
>
> The authors have submitted the response and also requested to discuss it with you.
>
> Could you please read their response and give your feedback? Thanks.
>
> Best,
>
> AC

---

### Author Response · Authors · 2023-11-18
**Summary Response to all Reviewers and AC**

We extend our gratitude to all the reviewers for their detailed and comprehensive reviews and time spent reviewing our manuscript.
We are delighted that the reviewers recognized our method's performance and appreciated our contributions.
We addressed their concerns in our responses respectively.

For the convenience of the reviewers and the AC, we summarize the modifications we did to our manuscript here:


1. **Adding experiments at higher bitrates**: We provided two extra points on Kodak and one extra point on audio at higher bitrates, to show that our method still works well at higher bitrates.


2. **Showing stability during training**: To illustrate the practicality and potential of our approach,  we added a plot of RECOMBINER’s training curve in Figure 10 in Appendix D.4,  showing that RECOMBINER’s training is robust and stable without requiring stabilization techniques.
As the figure shows, after beta adjusts to its appropriate level early in the training, the training curves essentially monotonically increase without any dips.


3. **Updating Baselines**: We updated Figure 3a to include ELIC and VTM as baselines on CIFAR-10 and Kodak. We find that they do not outperform RECOMBINER on CIFAR-10 at low bitrates, and our method is still SOTA in this region.  Note that we omit the ELIC curve on CIFAR-10. This is because we tested ELIC on CIFAR-10, but it did not work well on low-resolution images, with performance even worse than Balle 2018 and sometimes does not converge.
To better visualize the updated plot, we provide full-resolution and full-range rate-distortion curves on images in Figures 11 and 12 in the appendix.


4. **More comparison with VC-INR**: We have updated our paper to include VC-INR in our audio compression plot in Figure 3b and plan to include it for video compression in Figure 3c. Please see our response to Reviewer oN2Z for a detailed explanation.



5. Fixing typos and clarifying the sentences which may cause misunderstanding.

---

### Comment · Area_Chair_XUmL · 2023-11-23
**[ICLR 2024 Reviewers’ feedback] Please read authors’ responses and give your feedback**

Dear Reviewers,

Thanks again for your strong support and contribution as an ICLR 2024 reviewer.

Please check the response and other reviewers’ comments. You are encouraged to give authors your feedback after reading their responses. Thanks again for your help!

Best,

AC

---

### Meta-Review · Area_Chair_XUmL · 2023-12-13

**Metareview:**

The method is technically well-designed and well-motivated, especially, the hierarchical Bayes approach. Experimental results have verified the effectiveness of the proposed RECOMBINER method. The paper structure is clear and easy to follow. But, there are still some typos and unclear parts to be further explained. Overall, all reviewers gave positive ratings after the rebuttal.

**Justification For Why Not Higher Score:**

There are still some typos and unclear parts to be further explained.

**Justification For Why Not Lower Score:**

Overall, all reviewers gave positive ratings after the rebuttal.

---

### Decision · Program_Chairs · 2024-01-16

Accept (poster)